# Climate impacts from North American boreal forest fires

**Max J. van Gerrevink** [1] ✉, **Sander Veraverbeke** [1,2], **Sol Cooperdock** [3], **Stefano Potter** [3], **Qirui Zhong** [1,4], **Michael Moubarak**[5], **Anna-Maria Virkkala** [3,6], **Scott J. Goetz** [7], **Michelle C. Mack**[8], **James T. Randerson**[9], **Nick Schutgens** [1], **Merritt R. Turetsky** [10], **Guido R. van der Werf**[11] & **Brendan M. Rogers** [3]

The boreal forest biome is warming rapidly, impacting disturbance regimes and global climate. Boreal forest fires have intensified, initiating both climate warming (positive) and climate cooling (negative) impacts across spatial and temporal scales. Here we estimate climate impacts from boreal fires in Alaska and western Canada between 2001 and 2019 using integrated net radiative forcing metrics combining greenhouse gas and aerosol emissions from combustion, vegetation recovery, greenhouse gas emissions from fire-induced permafrost thaw and changes in surface albedo over a 70-year period. We find that fires across Alaska contributed, on average, to net climate warming ($0.35 \pm 4.66$ W m$^{-2}$ of burned area; one standard deviation), while fires across Canada contributed to net cooling ($-2.88 \pm 4.17$ W m$^{-2}$ of burned area; one standard deviation). Climate-warming fires occur preferentially in dry, high-elevation, steep permafrost landscapes with high pre-fire black spruce coverage and combust more carbon per unit area. Climate-cooling fires are driven by longer spring snow exposure and occur more frequently in continental regions near the treeline. This fine-scale characterization of component and net radiative forcing advances our understanding of the biogeophysical impacts of fires on high-latitude climate and highlights the need to prioritize fire management in carbon-rich permafrost regions to curb long-term warming.

The boreal biome is rapidly warming, affecting northern ecosystems and global climate[1,2]. Climate warming has altered fire regimes in boreal North America, leading to longer fire seasons and increasingly extreme and complex fire behaviour[3–8]. This trend is expected to persist during the twenty-first century with the potential to change forest composition, function, carbon stocks and ecosystem services[3,9–11]. Fire is a major natural disturbance mechanism across boreal North America, triggering both climate-warming (positive) and climate-cooling (negative) influences[11–14]. Greenhouse gas emissions from fires become well mixed in the atmosphere and persist for years to centuries, contributing to global-scale climate warming[15,16]. Fire aerosol emissions are short-lived and can lead to substantial regional warming or cooling depending on aerosol size and composition, surface albedo and interactions with clouds[13,17,18]. Typical stand-replacing crown fires across boreal North America[19] result in elevated surface albedo due to post-fire spring snow exposure lasting several decades, inducing a strong regional and seasonal cooling influence[12,16,20–22]. Regenerating forests are a carbon sink[23] as vegetation recovers and carbon sequestration occurs after a fire. Further, fires in permafrost landscapes can initiate or accelerate permafrost thaw and may therefore have a large, although uncertain, impact on the permafrost–carbon feedback[24].

---

**Fig. 1 | Cumulative mean climate radiative forcing from fires between 2001 and 2019 across Alaska and western Canada over a 70-year period. a**, Net radiative forcing map. **b**, Contribution of net radiative forcing (black circles), direct greenhouse gas and precursor emissions (red squares), permafrost greenhouse gas emissions (orange upside-down triangles), change in surface albedo (blue triangles), aerosol emissions (cyan hexagons) and vegetation recovery (green diamonds) for Interior Alaska and Boreal Shield fires. In total, $n = 2,259,662$ burned pixels were included for Interior Alaska and $n = 1,848,248$ for the Boreal Shield. The error bars indicate the uncertainty of each agent with one standard deviation of the mean based on the uncertainty assessment. **c**, Net radiative forcing over some selected fires in Interior Alaska. **d**, Net radiative forcing over some selected fires in central Canada. The ecoregion boundaries for Interior Alaska and the Boreal Shield are delineated by the dashed grey lines. Ecoregion data in **a,c,d** from the United States Environmental Protection Agency (https://www.epa.gov/eco-research/ecoregions-north-america). Basemaps in **a,c,d** from Natural Earth (https://www.naturalearthdata.com).

Although climate impacts from boreal forest fires display varying spatial and temporal dynamics, they can be directly compared using the metric of radiative forcing (W m$^{-2}$)[25–27], defined by the Intergovernmental Panel on Climate Change[28], as the change in net radiative flux at the top-of-atmosphere after allowing atmospheric temperatures to readjust to radiative equilibrium[28,29]. In this Article, we adapted the concept of radiative forcing to assess the net impact of North American boreal forest fires between 2001 and 2019 on global climate, compared with a counterfactual no-fire situation. We accounted for fire greenhouse gas and aerosol emissions, post-fire surface albedo change, vegetation recovery and fire-induced permafrost thaw (Extended Data Fig. 1 and Methods). Our radiative forcing estimates, in units of W m$^{-2}$ of burned area, capture climate impacts over a 70-year post-fire period (Supplementary Information section 1), using the 'middle-of-the-road' shared socioeconomic pathway SSP2–4.5[29].

Knowledge of the separate components and net radiative forcing from boreal forest fires could prove crucial for mitigating the impacts of boreal fires on climate through altered forest and fire management[30,31]. Previous estimates of radiative forcing from individual fire events in Interior Alaska and central Canada showed a net cooling[12,21] due to the dominant negative forcing from surface albedo changes relative to the warming effects from fire-induced greenhouse gas emissions. However, the total fuel consumption estimates from these individual fire events (1.76 ± 0.62 kgC m$^{-2}$ and 1.73 ± 0.37 kgC m$^{-2}$; uncertainties expressed as one standard deviation) were much lower than the average combustion from recent data synthesis and modelling estimates across central and western boreal North America of over 3 kgC m$^{-2}$ (refs. 19,32,33). This difference leads to lower estimated warming effects from greenhouse gas emissions for these fires than would be the case for a broader set of fires across the domain. In northwestern and western boreal North America, combustion per unit of burned

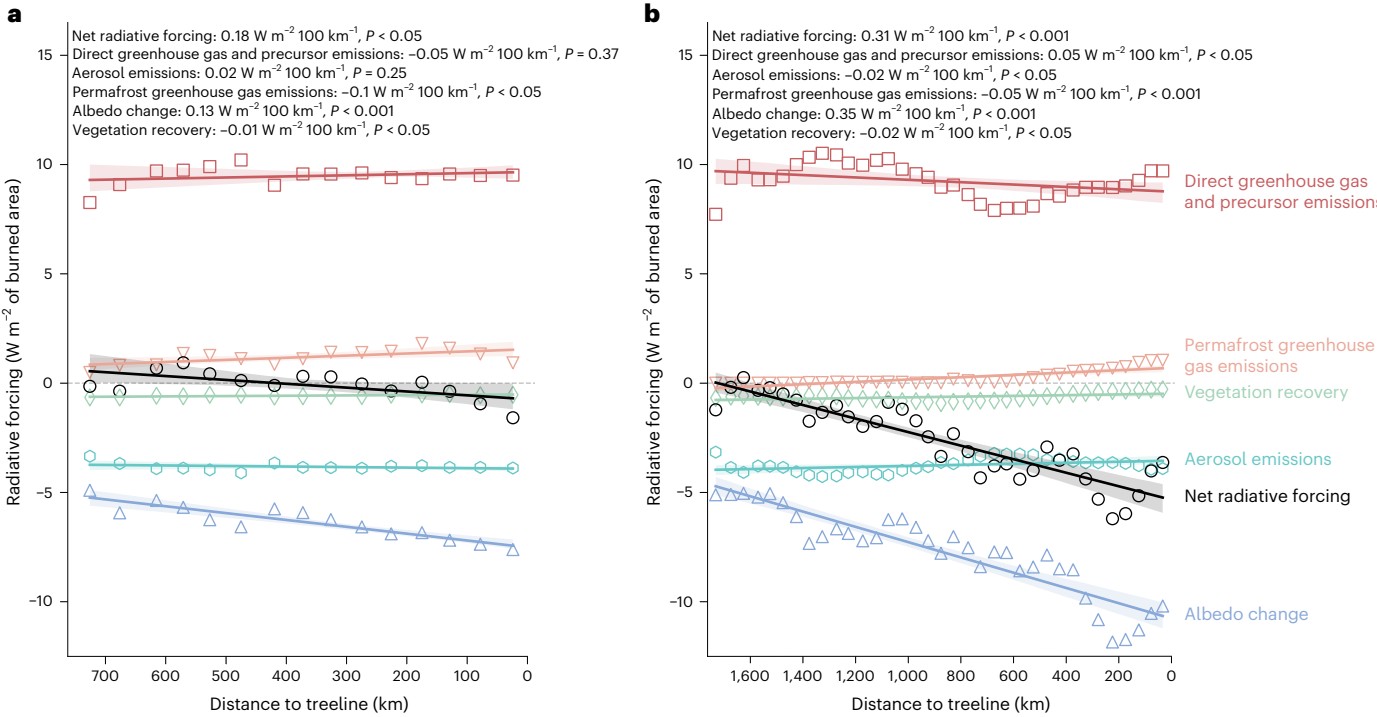

**Fig. 2 | Fire radiative forcing as a function of distance to the latitudinal treeline. a**, Alaska. **b**, Western Canada. Net radiative forcing (black circles), direct greenhouse gas and precursor emissions (red squares), permafrost greenhouse gas emissions (orange upside-down triangles), vegetation recovery (green diamonds), aerosol emissions (cyan hexagons) and change in surface albedo (blue triangles) trends in relation to distance to treeline are shown as mean values within 50 km intervals. The shading represents the 95% confidence interval around the regression line. Individual trends are shown with slopes in unites of W m$^{-2}$ 100 km$^{-1}$ and $P$ values. Statistical significance was determined by a two-sided $t$ test of the regression coefficient ($P < 0.05$, $P < 0.01$ and $P < 0.001$).

area (kgC m$^{-2}$) is comparatively higher due to deeper burning into organic soils, leading to more direct greenhouse gas emissions[32]. In addition, effects from greenhouse gas emissions due to fire-induced permafrost thaw, and aerosol indirect effects, were not accounted for in these estimates. Thus, while previous studies[12,21,34] have provided valuable insights into the governing climate impacts from boreal fires, they did not include some key forcing pathways and focused primarily on individual fires or a small number of events. Our holistic approach across western boreal North America takes a key step forwards in understanding the influence of boreal fires on climate. We present our data at the administrative level and further examine it at the ecoregion level to provide deeper insights into spatial variability and regional dynamics of how high-latitude fires impact climate.

## Spatial variability in climate impacts of high-latitude fires

Over a 70-year period, fires across western boreal North America had a net cooling effect, with a mean radiative forcing of −1.99 ± 4.29 W m$^{-2}$, where the uncertainty represents one standard deviation of the prediction uncertainty (Fig. 1 and Extended Data Table 1). This definition of uncertainty is applied throughout the manuscript unless stated otherwise. When not accounting for fire-induced permafrost greenhouse gas emissions, this cooling becomes more pronounced (−2.64 ± 4.28 W m$^{-2}$), highlighting the role of fire-induced permafrost thaw emissions. Regionally, Alaskan fires showed a net warming impact (0.35 ± 4.66 W m$^{-2}$) while Canadian fires contributed to a net cooling impact (−2.88 ± 4.17 W m$^{-2}$; Extended Data Table 1). In Alaska, 43% of burned areas resulted in net warming, compared with 10% in Canada. Among ecoregions, fires in Interior Alaska, Boreal Cordillera and Montane Cordillera on average showed a net climate-warming impact whereas fires across the Taiga Cordillera, Taiga Plain, Boreal Plain, Taiga Shield and Boreal Shield showed an average net climate-cooling impact (Extended Data Fig. 2).

To examine the influences of fire on climate across ecoregions, we highlight the Boreal Shield and Interior Alaska ecoregions, which show contrasting net climate impacts. On average, fires in the Boreal Shield showed a strong net climate-cooling influence of −4.23 ± 3.73 W m$^{-2}$ (Fig. 1d) while fires in Interior Alaska showed a net climate-warming influence of 0.34 ± 4.56 W m$^{-2}$ (Fig. 1c). The contrasting net climate impacts are embedded within ecosystem and landscape characteristics. The Boreal Shield ecoregion contains little to no permafrost, in contrast to Interior Alaska, which lies within the discontinuous permafrost zone (50–90% coverage)[35]. This biophysical difference results in a greater warming from fire-induced permafrost emissions in Interior Alaska (1.36 ± 0.71 W m$^{-2}$) than in the Boreal Shield (0.15 ± 0.08 W m$^{-2}$; Supplementary Fig. 11). In addition, post-fire changes in surface albedo led to a stronger climate-cooling impact in the Boreal Shield (−9.40 ± 1.51 W m$^{-2}$) compared with Interior Alaska (−6.48 ± 1.04 W m$^{-2}$), driven largely by differences in forest characteristics and prolonged snow exposure[36]. In the Boreal Shield, changes in post-fire surface albedo more than offset the warming caused by greenhouse gas and precursor emissions (that is, greenhouse gases emitted at the time of the fire; 7.88 ± 2.84 W m$^{-2}$) and from fire-induced permafrost thaw (0.15 ± 0.08 W m$^{-2}$). Amplified by the on-average cooling responses of aerosol emissions (−2.21 ± 1.88 W m$^{-2}$) and post-fire vegetation recovery (−0.66 ± 0.05 W m$^{-2}$), this results in net climate-cooling fires across the Boreal Shield ecoregion. By contrast, in Interior Alaska, climate cooling from post-fire changes in surface albedo only partially offset the climate-warming effects of the greenhouse gas and precursor emissions (9.32 ± 3.36 W m$^{-2}$). This residual climate warming along with additional permafrost greenhouse gas emissions offsets the climate cooling from aerosol emissions (−3.30 ± 2.81 W m$^{-2}$) and vegetation recovery (−0.56 ± 0.04 W m$^{-2}$), leading to, on average, net climate-warming fires.

We further examined the climate impacts of fires in Alaska and Canada per unit area burned in relation to the treeline[37]. The fires close

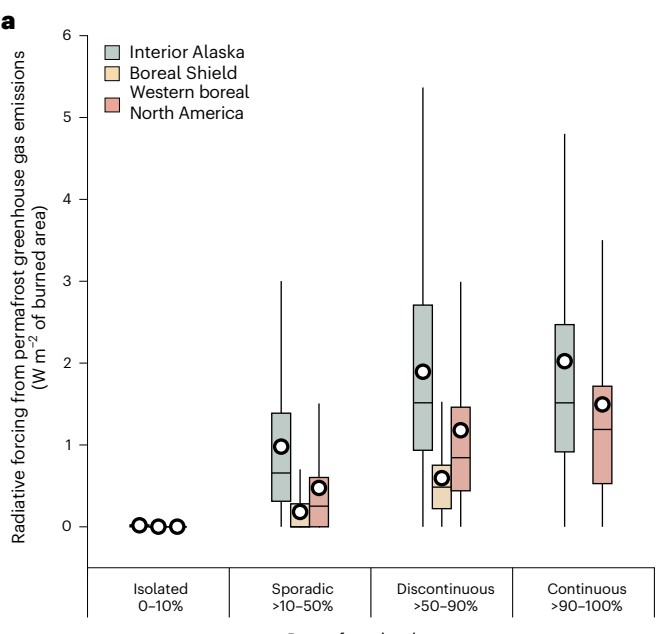

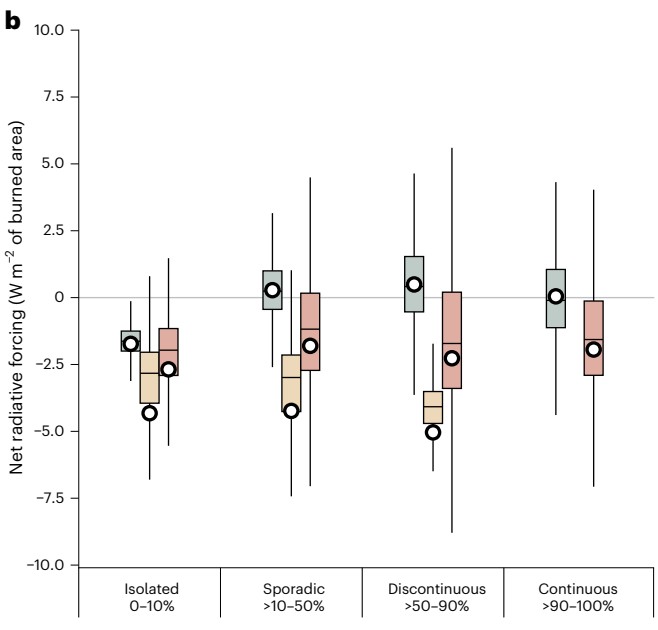

**Fig. 3 | Radiative forcing from permafrost greenhouse gas emissions induced by fire across landscapes with different permafrost extents. a**, Radiative forcing from permafrost greenhouse gas emissions under different permafrost landscapes from all fires across the study domain between 2001 and 2019. **b**, The net radiative forcing for different permafrost landscapes. Fires from the Interior Alaska are shown in green, Boreal Shield fires are in yellow, and domain-wide estimates across western boreal North America are in orange. Permafrost extent was defined using the permafrost zonation index from ref. 65. Burned pixels per permafrost landscape are as follows: Interior Alaska—isolated

($n = 6,897$), sporadic ($n = 1,303,778$), discontinuous ($n = 868,043$), continuous ($n = 80,944$); Boreal Shield—isolated ($n = 336,406$), sporadic ($n = 1,492,235$), discontinuous ($n = 19,607$), continuous ($n = 0$); western boreal North America—isolated ($n = 1,127,980$), sporadic ($n = 5,018,978$), discontinuous ($n = 3,631,818$), continuous ($n = 371,920$). Note, the Boreal Shield ecoregion does not have landscapes with continuous permafrost. Horizontal lines represent the median, the white circle represents the mean, and upper and lower limits of the boxes show the 25th and 75th percentiles. Whiskers extend up to 1.5 times the interquartile range.

to the treeline (up to 50 km south of the treeline), on average, showed a climate-cooling effect (Fig. 2). This effect is driven primarily by the strong post-fire increases in surface albedo that outweigh the influence of direct greenhouse gas emissions from combustion and fire-induced permafrost thaw. However, there are large regional differences in radiative forcing from changes in surface albedo. Fires near the treeline in Canada experience approximately 34% more cooling from changes in surface albedo ($-10.30 \pm 4.56$ W m$^{-2}$, one standard deviation of spatial variability) compared with fires near the treeline in Alaska ($-7.69 \pm 2.33$ W m$^{-2}$, one standard deviation of spatial variability). Further south of the treeline, in areas with earlier spring snowmelt, the cooling impact from prolonged snow exposure diminishes. Canadian fires more than 1,000 km south of the treeline result in up to 30–50% less cooling from changes in surface albedo than those near the treeline.

Across western boreal North America, permafrost thaw plays an important role for climate-warming fires. Our first-order estimates suggest that fire-induced greenhouse gas emissions from permafrost active layer thickening contribute to an average positive forcing of $0.65 \pm 0.34$ W m$^{-2}$. Domain-wide, fires occurring in sporadic permafrost landscapes (10–50% coverage) show an average permafrost radiative forcing of $0.47 \pm 0.25$ W m$^{-2}$. These fires constitute 49% of the burned area between 2001 and 2019. For fires in the discontinuous permafrost zone (50–90% coverage, 35% of burned area), this increases to $1.18 \pm 0.63$ W m$^{-2}$, whereas fires in the continuous permafrost zone (≥90% coverage, 4% of burned area) result in a forcing of $1.50 \pm 0.80$ W m$^{-2}$ (Fig. 3). In large parts of the Interior Alaska, Montane Cordillera and Boreal Cordillera ecoregions, higher fuel consumption and thus emissions of greenhouse gases and precursors offset the albedo-driven cooling, leading to climate-warming fires (Extended Data Fig. 2). Notably, the absence of permafrost in the Montane Cordillera and the southern part of the Boreal Cordillera further distinguishes their fire dynamics and associated climate impacts from

those in other ecoregions. In Interior Alaska, our findings highlight a substantial role for greenhouse gas emissions from fire-induced permafrost thaw. Without accounting for post-fire greenhouse gas emissions from permafrost thaw, Alaskan fires would result in an average climate-cooling effect of $-1.05 \pm 4.60$ W m$^{-2}$ (Extended Data Table 1).

## Landscape, forest and fire characteristics of warming fires

The net radiative forcing of boreal fires is strongly influenced by fire and landscape characteristics. Using a large field database[38], we found, on average, climate-warming fires occurred in drier landscapes ($P = 0.05$), on steeper slopes ($P < 0.001$) and at higher elevations ($P < 0.001$; Table 1). Pre-fire forest characteristics, such as stand density and species composition, are drivers of fire severity due to their close association with fuel availability[39,40]. Notably, net-warming fires occurred in areas with a higher percentage of pre-fire black spruce ($P < 0.001$) while tree density ($P = 0.81$) and age ($P = 0.21$) did not show significant differences between climate-warming and climate-cooling fires. Nevertheless, landscape features, including vegetation cover, soil properties and topography, influence fire behaviour. We analysed fire characteristics such as fire size, day of burning, burn depth and total carbon losses using the government fire polygons and the Arctic–Boreal Vulnerability Experiment fire emissions database (ABoVE-FED)[41]. We found that, on average, climate-warming fires—comprising 21% of fires across boreal North America—were generally not larger in size yet combusted more carbon per unit area. On average, climate-warming fires consumed $3.49 \pm 0.38$ kgC m$^{-2}$, compared with $2.81 \pm 0.76$ kgC m$^{-2}$ for climate-cooling fires ($P < 0.001$) and burned deeper into the organic soil layer ($11.6 \pm 1.7$ cm versus $9.6 \pm 2.3$ cm; $P < 0.001$) (Extended Data Fig. 3). Consequently, climate-warming fires combusted 50% of pre-fire soil organic carbon compared with 39% for climate-cooling fires. The difference in combustion reflects landscape and hydrological characteristics

**Table 1 | Landscape and fire characteristics of climate-warming and climate-cooling fires**

| Variable | Climate-warming fires (±s.d.) | Climate-cooling fires (±s.d.) | P value | Sample size (W)\|(C) |
|---|---|---|---|---|
| Net radiative forcing | 0.84 (±0.61) W m$^{-2}$ | −3.37 (±4.06) W m$^{-2}$ | | All fires |
| Moisture class[a] | Subxeric | Subhygric | 0.05* | (138)\|(476) |
| Slope | 6.20 (±7.19)° | 2.23 (±5.01)° | <0.001* | (134)\|(269) |
| Elevation | 553 (±236) m | 322 (±126) m | <0.001* | (149)\|(271) |
| Stand density | 0.67 (±0.63) stems m$^{-2}$ | 0.69 (±0.71) stems m$^{-2}$ | 0.81 | (99)\|(414) |
| Stand age | 103 (±51) years | 96 (±49) years | 0.21 | (134)\|(344) |
| Fraction black spruce (Pre-fire) | 0.85 (±0.28) | 0.71 (±0.36) | <0.001* | (151)\|(402) |
| Total C combustion (ABoVE-FED) | 3.49 (±0.38) kg C m$^{-2}$ | 2.81 (±0.76) kg C m$^{-2}$ | <0.001* | All fires |
| Burn depth (ABoVE-FED) | 11.6 (±1.7) cm | 9.6 (±2.3) cm | <0.001* | All fires |
| Fire size (long-term databases) | 6.62 (±21.6)×1,000 ha | 7.01 (±27.3)×1,000 ha | 0.51 | All fires |
| Day of burning (long-term databases) | 199 (±48) day of year | 187 (±37) day of year | <0.001* | All fires |

s.d. indicates standard deviation of the mean; W and C represent the sample sizes of climate-warming and climate-cooling fires, respectively; the total number of 'All fires' is 11,795. [a]The variable moisture class is a categorical dataset. We present the mode of climate-warming and climate-cooling fires in this case. The P value of the moisture class was assessed with a two-sided Mann–Whitney U test. All other P values were evaluated using two-sided Welch's t test. The variables moisture class, slope, elevation, stand density, stand age and fraction black spruce (pre-fire) are pixel-based and are derived from the ABoVE synthesis field dataset of combustion measurements[38]. The variables total C combusted, burn depth, fire size and day of burning are fire perimeter-based and are derived from the ABoVE-FED[41] and long-term governmental fire databaseka and Canada[15,63,64]. References for data sources are given in Methods. Exact P values are provided whenever possible. *P<0.05.

that promote deeper and more severe burning and larger carbon release per unit area. On average, climate-cooling fires ignited earlier in the year, burning on average 12 days earlier than climate-warming fires (day of burning of climate-warming fires 199 ± 48 versus climate-cooling fires 187 ± 37; P < 0.001; Table 1). Pre-fire forest type also influences fires' net climate radiative forcing (Extended Data Fig. 6b). Although fires in both evergreen needleleaf and deciduous broadleaf forests resulted in an on-average net climate-cooling effect (−0.70 ± 2.39 W m$^{-2}$ and −1.24 ± 1.55 W m$^{-2}$, respectively; standard deviation expressed as spatial variability), 40% of burned areas in evergreen needleleaf forests resulted in net climate warming, compared with 12% in deciduous forests. On average, evergreen needleleaf forests showed 34% less climate cooling from vegetation recovery compared with deciduous broadleaf forests, while the climate warming from permafrost thaw emissions, on average, was 76% higher in evergreen needleleaf forests. The difference in net climate radiative forcing between evergreen needleleaf and deciduous broadleaf stands is driven partly by fire-induced permafrost thaw emissions, which account for 23% of the absolute contribution to the overall difference (Extended Data Fig. 4). The pattern reflects vegetation traits as evergreen needleleaf forests accumulate thicker organic soils and exhibit longer recovery rates, whereas deciduous broadleaf forests promote faster carbon turnover and shallower permafrost, limiting post-fire warming impacts[10].

## Implications and limitations

Here we document climate impacts from western boreal North American fires with an unmatched spatial resolution of 500 m. Our findings provide valuable insights into understanding the impacts of fires on climate in a region that has recently experienced record-breaking fire seasons[7,42]. We found, on average, Alaskan fires have a net climate-warming impact and Canadian fires a net climate-cooling impact (Extended Data Table 1). The confidence levels associated with the sign of our net radiative forcing estimates are between 66 and 80% on average, at the fire level (Extended Data Fig. 5). Climate-warming fires were found in ecoregions with higher fuel consumption[32] and in ecoregions with greater permafrost extent. Climate-cooling fires were found in non-permafrost landscapes and ecosystems characterized by lower pre-fire dominance of black spruce. Similar to previous work done at individual field sites[12,21], the cooling influence of post-fire surface albedo changes in such ecosystems exceeded the warming effects from greenhouse gases emitted at the time of fire and from permafrost thaw.

The radiative forcing agents presented in this study have different spatial and temporal footprints, which is important to consider. For example, climate radiative forcing from fire aerosol emissions is regional and short-lived, on the order of days to weeks. The negative radiative forcing from changes in post-fire surface albedo, which is the primary agent leading to climate-cooling fires in our assessment, lasts decades but operates regionally and primarily during the late winter and early spring[20,43]. By contrast, the climate warming caused by greenhouse gas emissions from fire is long-lived and distributed globally[44,45], influencing global temperatures and impacting policy-relevant metrics such as humanity's allowable carbon emissions to stay within Paris-aligned global temperature thresholds (Extended Data Fig. 6). Hence, implementing strategies such as targeted forest and fire management in regions dominated by fires linked to climate warming or focusing efforts across landscapes with vulnerable, carbon-rich permafrost, could help mitigate the cascading effects of climate-warming fires. Fire management across boreal North America can alter natural fire regimes and increase fuel accumulation. However, targeted suppression, prescribed burning and forest management may reduce the frequency and intensity of climate-warming fires, thereby reducing greenhouse gas emissions and limiting permafrost thaw. In addition, restoring fire-adapted ecosystems can enhance the resilience of these landscapes while maintaining natural fire dynamics.

Our results highlight the underlying role of fire-induced permafrost thaw in areas with climate-warming fires. The radiative forcing estimates we present on permafrost thaw are dependent on the spatial distribution of permafrost extent and near-surface permafrost carbon stored in the thickened active layer after fire[46]. Consequently, our first-order permafrost radiative forcing estimates have a larger magnitude in the northernmost ecoregions of our study area. Differences between ecoregions arise due primarily to variations in near-surface carbon stocks and permafrost extent, with southern ecoregions generally having lower carbon densities and more isolated permafrost pockets, resulting in a smaller permafrost radiative forcing (Fig. 3 Supplementary Fig. 11). Our estimates of post-fire permafrost emissions do not incorporate post-fire subsidence and abrupt thaw events due to data and knowledge limitations, which may lead to conservative estimates for ice-rich permafrost regions. Abrupt thaw events such as thermokarst are confined primarily to ice-rich permafrost landscapes, which represent roughly a third of the pan-boreal region[47,48]. Abrupt thaw events are hotspots for carbon emissions and can be triggered by single-season events such as fires[49–51]. Carbon

emissions through abrupt permafrost thaw are estimated to roughly double the radiative impact of permafrost emissions from gradual thaw alone due to the higher $CH_4$ emission ratios associated with abrupt thaw[49]. Permafrost thaw is also closely coupled with vegetation dynamics, influencing maximum thaw depth, rooting depth and water availability[52,53]. Permafrost thaw may alter post-fire vegetation composition and structure, influencing canopy shading effects and changing soil exposure to solar radiation[52,53]. Future work should prioritize quantifying post-fire permafrost carbon emissions using in situ measurements to better constrain post-fire carbon releases from permafrost carbon[54,55]. We assumed here that the post-fire carbon dynamics associated with vegetation composition changes following permafrost thaw are captured in our analysis of post-fire vegetation recovery estimates. We recognize that we did not explicitly account for potential ecosystem shifts after fire-induced permafrost thaw, which may alter allocation between above- and belowground carbon pools[10,56] and further contribute to surface energy budget changes. While explicit inclusion of successional trajectories could lead to further improvements in the post-fire vegetation recovery component of our framework, the potential effect on the net radiative forcing estimates would probably be negligible given the small contribution of the post-fire vegetation recovery component to the net forcing estimates[9]. Nevertheless, we chose to model the permafrost emissions and net ecosystem exchange (NEE) of $CO_2$ separately as there are indications that the NEE product that we used misses an important source of carbon emissions in post-fire environments[57]. In addition, the NEE model is dependent on data from only a few burned sites in permafrost terrain, and therefore its sensitivity to adequately capture permafrost processes and fluxes may be limited. Future work would benefit from conceptually bringing together carbon dynamics and fluxes from post-fire recovering vegetation and soil respiration from modern plant and aged permafrost carbon.

The future climate-cooling impact from boreal fires due to increased spring snow albedo is expected to diminish because of reduced snowfall and earlier snowmelt resulting from climate change[58]. This shift could lead to transitions in regions where the net climate radiative forcing of fires is dominated by surface albedo changes, such as the Boreal Plain and Boreal Shield ecoregions, potentially shifting fires in these regions towards a climate-warming state. More frequent fires will probably impact post-fire succession by enhancing the exposure of mineral soil seed beds, thereby promoting deciduous stand replacement[10,23,59–61]. Losses in surface and deep permafrost soil carbon may be offset by more aboveground carbon sequestration across mixed and deciduous stands[10]. However, longer growing seasons may result in longer fire seasons and therefore more burned area. Intensification of fire regimes may shorten the carbon residence time on land, transforming boreal forests from a carbon sink to a carbon source[62]. Understanding and addressing these changes will be critical for managing boreal ecosystems and mitigating their climate impacts.

## Online content

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

¹Faculty of Science, Vrije Universiteit Amsterdam, Amsterdam, the Netherlands. ²School of Environmental Sciences, University of East Anglia, Norwich, UK. ³Woodwell Climate Research Center, Falmouth, MA, USA. ⁴College of Urban and Environmental Sciences, Peking University, Beijing, China. ⁵Hamilton College, Clinton, NY, USA. ⁶Finnish Meteorological Institute, Helsinki, Finland. ⁷School of Informatics, Computing, and Cyber Systems, Northern Arizona University, Flagstaff, AZ, USA. ⁸Center for Ecosystem Science and Society and Department of Biological Sciences, Northern Arizona University, Flagstaff, AZ, USA. ⁹Department of Earth System Science, University of California, Irvine, CA, USA. ¹⁰Renewable and Sustainable Energy Institute, Department of Ecology and Evolutionary Biology, University of Colorado, Boulder, CO, USA. ¹¹Meteorology & Air Quality Group, Wageningen University and Research, Wageningen, the Netherlands. ✉e-mail: m.j.van.gerrevink@vu.nl

## Methods

We estimated the net radiative forcing impacts from North American boreal forest fires between 2001 and 2019 relative to a no-fire baseline. Our estimates are expressed in units of W m$^{-2}$ of burned area and represent the integrated climate impact over a 70-year post-fire period under the shared socioeconomic pathway SSP2–4.5[29]. We used burned area data from the ABoVE-FED[32] and quantified the net climate radiative forcing impacts at 500 m resolution. In this work, net radiative forcing represents the balance of multiple fire-driven climate forcing components, integrating greenhouse gas and aerosol emissions from combustion, changes in surface albedo, post-fire vegetation recovery and greenhouse gas emissions associated with fire-induced permafrost thaw. We measured these metrics in relation to the treeline to capture spatial and ecological variation in fire-driven radiative forcing across boreal forests. Finally, we reported landscape and forest characteristics at the pixel level and fire-specific attributes at the fire level to examine their influence on the net climate radiative forcing.

### Radiative forcing from greenhouse gases, precursors and aerosol emissions

We used gridded estimates of direct fire carbon emissions from the ABoVE-FED together with published emission factors to estimate the climate impacts from greenhouse gas and aerosol emissions[28,32,41,66–68] (Supplementary Table 1). The ABoVE-FED provided annual gridded estimates of burned area and carbon emissions across Alaska and Canada at 500 m spatial resolution between 2001 and 2019[41]. These estimates were modelled using machine learning based on in situ observations from ref. [38] and geospatial products of fire severity, fire weather, climate, topography, land cover and soils.

We computed the radiative forcing from well-mixed greenhouse gases $CO_2$, $CH_4$ and $N_2O$ using simplified radiative expressions based on previous studies[13,25,26,69–71] (Supplementary Information equations (1)–(5) and section 2.1) following ref. [72]. To account for the variable $CO_2$ lifetime under SSP2–4.5, we derived concentration-based impulse–response functions under climate change[71] (Supplementary Fig. 1 and Supplementary Information section 2.1). Airborne fractions of $CH_4$ and $N_2O$ were lifetime dependent. We assumed a fixed atmospheric lifetime of 12.4 years for perturbations to $CH_4$ (ref. [27]). For $N_2O$, we accounted for the feedback mechanism of future $N_2O$ concentrations on its own lifetime[73]. The radiative forcing from $O_3$ emissions was calculated as a function of fire-related CO emissions[69,70,72]. Climate forcing from ozone precursors, $NO_x$, CO, non-methane volatile organic compounds and aerosols (organic and black carbon), were incorporated following a time-integrated radiative forcing method based on global warming potentials (GWPs)[67,68,72,74,75]. We used the climate-warming effects of precursors at 20 years and 100 years relative to $CH_4$ to derive radiative forcing estimates for $NO_x$, CO and non-methane volatile organic compounds (Supplementary Information equation (6) and section 2.1). Aerosol emissions are usually removed from the atmosphere within weeks after a fire event by means of dry and wet deposition. To estimate the direct radiative forcing of organic and black aerosols, we used a time-integrated method based on their GWPs at 20 years. Because GWP values do not directly correspond to radiative forcing, we derived a conversion factor by comparing the GWP from organic and black carbon with that of $CH_4$, for which radiative forcing estimates are available (Supplementary Information equation (6) and section 2.2)[72]. This yielded a ratio that we assumed also applies to radiative forcing at 20 years post-fire; however, due to the short-lived nature of aerosol impacts, we assumed that the total aerosol forcing occurred in the first year following the fire. We incorporated indirect radiative forcing from aerosols by multiplying the direct radiative forcing estimates from aerosols with the ratio of indirect to all-sky direct radiative forcing from aerosol emissions[13] (Supplementary Information section 2.2).

### Radiative forcing from post-fire surface albedo changes

To capture the decadal impacts of fire on land surface albedo, we implemented a retrospective space-for-time approach using historical fire events from the Alaska Large Fire Database[64] and Canadian National Fire Database[15,63]. This method allows us to infer long-term post-fire albedo dynamics by using spatially distributed fires of different ages as proxies for temporal changes. Specifically, we combined historical fire records with the satellite-derived surface albedo estimates to reconstruct how albedo evolves after a fire within a space-for-time approach. For example, if a fire was recorded in 1960, we can examine satellite-retrieved surface albedo for that burned area in June 2010. The albedo observed in this image represents conditions 50 years after the fire. By systematically applying this approach across fires of different ages, we constructed a time series of post-fire surface albedo spanning multiple decades.

We used random forest regressor models[76] to model post-fire changes in surface albedo with constrained observations of the Moderate Resolution Imaging Spectroradiometer (MODIS) albedo product[77]. Satellite products currently model albedo under two extreme conditions: pure diffuse illumination (known as white-sky albedo) and pure direct illumination (known as black-sky albedo). These albedo values are derived from multiple angle measurements of surface reflectance using the RossThick-LiSparse Reciprocal kernel-driven model, with Bidirectional Reflectance Distribution Function[78] parameters specific to each location and time. However, to better represent actual ground illumination conditions, we used a daily blue-sky short-wave albedo product[77]. Blue-sky albedo combines both diffuse and direct illumination and is calculated using RossThick-LiSparse Reciprocal kernel equations with adjusted coefficients reflecting atmospheric and view-geometry conditions. We used the MODIS albedo product over Landsat-derived estimates because MODIS provides consistent daily observations with a spectral resolution well suited for robustly quantifying post-fire surface albedo. By contrast, Landsat's sparse temporal coverage and spectral bands limit its ability to capture gradual, landscape-scale albedo changes relevant for climate radiative forcing calculations.

We established a monthly mean blue-sky albedo for January through December filtered by various quality flags (Supplementary Information section 2.3)[20] and used this as the target variable for the random forest models. We trained separate random forest regressor models for each month and then that month within each post-fire year using a suite of historical environmental and climatological features. We incorporated environmental features related to permafrost[65], soil[79] and topography. Soil properties were acquired from SoilGrids[79] and integrated to represent the topsoil up to 30 cm deep. High-spatial-resolution climate data were extracted from ClimateNA[80,81]. ClimateNA uses high-resolution Parameter-elevation Regressions on Independent Slopes Model and WorldClim Global Climate data to represent present-day climate variables. ClimateNA uses data from the Coupled Model Intercomparison Project phase 6, which is downscaled to include future projections at 1 km resolution across North America[80]. Climate projections are provided as long-term bioclimatic variables in distinctive 30-year epochs using an ensemble mean of 13 different atmosphere–ocean general circulation models. We interpolated each climate variable between epochs on an annual time step using cubic spline interpolation techniques, with climate data from the 1980–2010 epoch serving as the reference point. All bioclimatic and biophysical variables were resampled using nearest neighbour interpolation techniques for all pixels to match our grid with 500 m spatial resolution. Pre-fire monthly mean values of albedo were based on all observations available within a given month in the MODIS record.

Although random forest regressor models do not require collinear variables to be removed, we applied feature selection to simplify each model by eliminating features outside of the top ten most influential

features according to their overall mean permutation feature importance across all models (Supplementary Table 2). After feature elimination, the retained predictors for surface albedo included permafrost extent, ruggedness, elevation, soil pH, silt percentage, climate moisture deficit index, evaporation, autumn precipitation and temperature in autumn and spring. Permafrost extent and soil texture influence vegetation type, which in turn affects reflectivity. Rugged terrain and elevation influence both snow accumulation and persistence while seasonal temperature and precipitation patterns may regulate vegetation growth, surface moisture and snow cover (Supplementary Fig. 3). We cross-validated each model by retaining 20% of the data for testing. We used $R^2$ and root mean squared error values as primary model performance metrics, calculated on the testing set averaged across all 70 post-fire years for a given month. The predictive performance demonstrated strong correlations between observations and predictions of post-fire surface albedo ($R^2$ ranges between 0.78 and 0.94 with root mean squared errors between 0.01 and 0.06; Supplementary Fig. 2). We do not explicitly account for spatial autocorrelation in our models. As a result, our model performance metrics may be slightly inflated due to the spatial clustering of environmental conditions. Nonetheless, our model performances are very similar to those reported by Potter et al.[20], who explicitly accounted for spatial autocorrelation. This similarity is expected given that we used the same target variable, have an overlapping spatial domain and relied on comparable features describing bioclimatic, vegetation and soils characteristics. This suggests that any potential inflation caused by spatial autocorrelation is minor and does not compromise the robustness of our models. However, by splitting the data according to month and year, we aim to minimize the influence of spatial dependencies over the time.

We converted post-fire surface albedo predictions into radiative forcing estimates using all-sky monthly spatially distributed specific albedo radiative forcing kernels[82] that provide top-of-atmosphere forcing rates per unit change in surface albedo. The original spatial resolution of 0.25° of the kernels was resampled to our 500 m spatial grid using nearest neighbour interpolation, aligning with the spatial resolution of the albedo product. By doing so, we computed pixel-based monthly radiative forcing estimates by multiplying the albedo radiative forcing kernel with fire-induced changes in albedo. The monthly radiative forcing estimates were then summed to create yearly post-fire radiative forcing estimates at a 500 m resolution.

### Radiative forcing from post-fire carbon sequestration

We derived spatially explicit post-fire $CO_2$ uptake curves using random forest regression models. These models were trained using the same retrospective space-for-time approach as for the post-fire surface albedo predictions. In this case, we integrated historical fire records with an upscaled NEE of $CO_2$ dataset. The retrospective space-for-time approach in this case reconstructs the trajectory of carbon dynamics following fire over multiple decades. The NEE dataset consists of monthly gridded estimates of $CO_2$ flux (gC m$^{-2}$ month$^{-1}$) at a 1 km spatial resolution from 2001 to 2020[57,83]. This monthly NEE product was upscaled from terrestrial eddy covariance and chamber flux observations using machine-learning techniques, which incorporated a wide range of geospatial variables, including climate, vegetation, topography and soils[83].

We calculated annual NEE budgets for 2001–2019 from the monthly gridded estimates, and resampled pixels to match with our 500 m grid using nearest neighbour resampling. In our retrospective space-for-time approach, if a fire was recorded in 1980, we can examine the post-fire carbon dynamics for that burned area in, for example, 2010. In this example, the annual NEE estimates across this area represent conditions 30 years after the fire. Similar to our post-fire albedo predictions, we trained separate random forest regressor models for each post-fire year, with annual NEE as the target variable, using a suite of environmental and climatological variables (Supplementary Table 2).

Unlike the monthly surface albedo models, we developed annual NEE models as we are interested primarily in yearly post-fire carbon flux impacts. We used feature elimination to simplify our models according to their overall mean permutation feature importance across all models. After feature elimination, the retained predictors for post-fire NEE included ruggedness, degree-days above 18 °C, extreme minimum temperature, evaporation and seasonal climate variables such as precipitation in spring and autumn, temperature in autumn, winter, and spring, and a measure of continentality (Supplementary Fig. 5). Both temperature and precipitation influence microbial activity and vegetation growth while evaporation reflects water availability. Terrain ruggedness impacts the distribution of vegetation and establishment of microclimates, while the measure of continentality, here defined as the difference in temperature between the mean coldest and warmest months of the year, accounts for seasonal temperature extremes, which shape ecosystem productivity. The feature importance analysis shows the relative contribution of each predictor assessed across all 70 post-fire years (Supplementary Fig. 5).

Each model was trained separately for post-fire years using an 80/20 train-test split, and performance was evaluated using $R^2$ and root mean squared error. The cross-validated models yielded very strong fits with $R^2$ ranging between 0.86 and 0.95 and root mean squared errors varying between 9.52 and 13.26 gC m$^{-2}$ year$^{-1}$ (Supplementary Fig. 4). We derived temporally and spatially explicit NEE dynamics for a 70-year post-fire period using these random forest regressors. We do not directly address spatial autocorrelation in our model; hence, performance metrics may be slightly inflated due to the spatial clustering. However, by splitting our data in the time dimension as part of the training-validation procedure, we reduce the potential effects of temporal dependencies. This approach, similar to how we handle the changes in post-fire surface albedo, reduces temporal autocorrelation effects in our models.

Yearly NEE dynamics were converted into radiative forcing estimates using the simplified radiative forcing expression for $CO_2$ (Supplementary Information equations (1)–(5) and section 2.1). In doing so, we accounted for the direct carbon emissions from the fire that affect post-fire atmospheric concentration to isolate the influence from post-fire carbon uptake. The total radiative forcing from post-fire vegetation recovery over the 70-year period is calculated as the cumulative sum of annual carbon uptake throughout this time frame.

It is important to note that our approach for post-fire vegetation recovery has certain limitations, particularly in capturing the full scope of permafrost carbon emissions. The annual NEE budgets we derive from Virkkala et al.[57] demonstrate a tendency to underestimate carbon sources in burned areas. One potential reason for the underestimation of $CO_2$ sources at burned sites is the omission of fire-induced permafrost thaw. While the model may capture some permafrost influences through site data and environmental correlations, it lacks a mechanistic representation of thaw processes and their specific impacts on carbon fluxes. The annual NEE budgets can underestimate carbon release, with biases reaching up to 75 gC m$^{-2}$ month$^{-1}$ at burned sites during cross-validation, with an average of 5.2 ± 23.1 gC m$^{-2}$ month$^{-1}$ (ref. 57). This documented bias is approximately half of the reported permafrost carbon emissions from active layer thickening after fire[57]. However, our estimates are conservative as they do not account for post-fire subsidence or abrupt thaw following fire. Carbon emissions from abrupt thaw are estimated to roughly double the radiative impact of permafrost emissions from gradual thaw[49].

### Radiative forcing from fire-induced permafrost thaw

We included estimates of the permafrost–carbon feedback following fires driven by active layer thickening by combining carbon release curves after thaw with soil organic carbon profiles, spatially explicit permafrost probability, a seasonality factor and changes in active layer thickness[51,65,84,85]. Annual carbon release curves after thaw were derived

from lab experiments[85] (Supplementary Fig. 6b). The carbon release curves were derived from ref. 85, which presented cumulative carbon release as a percentage of total carbon up to 10 years under aerobic conditions at a constant temperature of 5 °C (Supplementary Information section 2.5). We used the annual carbon release, as a function of incubation time, as a proxy for annual carbon release for time since the fire. However, as boreal temperatures are not consistently 5 °C, we applied a seasonality factor based on the number of frost-free days in each post-fire year to refine our estimates. This correction factor was calculated in a spatially and temporally explicit manner and represents the boreal growing season, during which conditions for permafrost–carbon decomposition are present. To achieve this, we used the number of frost-free days from ref. 80 and divided this by the total number of days in each post-fire year to obtain a fraction representing the relative length of the growing season. Natali et al.[86] showed that $CO_2$ release in controlled laboratory settings is nearly three times more sensitive to temperature changes than what is observed in natural field conditions. Laboratory incubations isolate the effect of temperature on microbial respiration, minimizing the influence of other environmental factors such as soil moisture. Therefore, the controlled conditions in the lab may lead to an overestimation of temperature sensitivity as they do not capture the buffering effects of other drivers that regulate $CO_2$ emissions from permafrost soils. Hence, to account for the higher temperature sensitivity observed in controlled laboratory settings compared with in situ field measurements, we applied a scaling factor of 2.93 (ref. 86). We validated the use of this scaling factor by comparing our modelled daily $CO_2$ fluxes with in situ rates of aged carbon release as $CO_2$ for a 6- (ref. 55) and 9- (ref. 54) year post-fire environment (Supplementary Information section 2.5 and Supplementary Fig. 7). The use of the scaling factor provides a more realistic estimate of fire-induced permafrost $CO_2$ emissions under natural conditions. Using the dataset from ref. 85, we created separate carbon release curves for mineral soils (<20% C) and organic soils (≥20% C), differentiating them according to soil organic carbon content derived from SoilGrids[79]. We used the active layer thickness dataset from the European Space Agency Climate Change Initiative on permafrost to derive the pre-fire active layer thickness (ALT) from the year before the fire and acquired post-fire fractional changes in ALT (dALT) from a previous literature review[51,87] (Supplementary Fig. 6a). We used the pixel-based zonation index from ref. 65 to delineate the probability of permafrost within each burned pixel. While this index provides valuable insights into the permafrost distribution, it does not account for vegetation–permafrost feedbacks, which could influence the thermal properties and stability of the permafrost in certain regions. Given the probability of permafrost, dALT and pre-fire ALT, we estimated the fraction of the soil column that is representative for the new active layer due to fire-induced thaw within each pixel and the proportion of the soil organic carbon (SOC) stock that becomes vulnerable to decomposition after the fire. We reconstructed SOC profiles up to 3 m (intervals: 0–30 cm, 30–50 cm, 50–100 cm, 100–200 cm and 200–300 cm) using data from ref. 84. However, since detailed information on the exact distribution of SOC within each interval is not available, we assumed an equal distribution of carbon with depth within a given interval. For example, if the pre-fire ALT at a given location is 70 cm and the post-fire ALT increases to 110 cm, this means that an additional 40 cm of soil carbon becomes vulnerable to decomposition. Of this, 30 cm falls within the 50–100 cm interval, which accounts for 60% of the SOC in the 50–100 cm interval, while the remaining 10 cm that extends into the 100–200 cm interval accounts for only 10% of the SOC in the 100–200 cm interval. This depth-based distribution estimate determines how much carbon is newly exposed to decomposition processes from fire-induced permafrost thaw. In addition, we approximated the partitioning of carbon emissions into $CO_2$ and $CH_4$ emissions. We used the average $CH_4$/C ratio of 15.89% from ref. 88 to calculate the proportion of carbon that will be emitted as $CH_4$ while assuming that the remaining carbon will be emitted as $CO_2$.

We estimated pixel-based $CO_2$ and $CH_4$ emissions by combining carbon release curves with the proportion of SOC vulnerable to thaw after the fire and the spatially explicit seasonality factor that represents the relative length of the growing season at this location. Our pixel-based permafrost emissions estimates were converted into radiative forcing using the simplified radiative forcing expressions[26] (Supplementary Information equations (1)–(5) and section 2.1). The total permafrost radiative forcing is presented as combined impacts of $CO_2$ and $CH_4$.

**Landscape, forest and fire characteristics and treeline data**
We used the ABoVE synthesis field dataset of combustion measurements[38], spanning six ecoregions, to analyse differences between fires that contributed to climate warming and those that contributed to climate cooling. This dataset captures a broad range of topographical variables, pre-fire stand characteristics and alterations in carbon pools from various fires between 2004 and 2015. We grouped all fire pixels into categories according to their climate feedback, either climate warming or climate cooling, across various ecoregions. This grouping was done to ensure sufficient sample size for our analysis. By doing so, we assessed the statistical differences in the means of climate-warming and climate-cooling fires across landscape characteristics such as forest density, stand age, species composition and landscape position and their distributions for climate-warming and climate-cooling fires using Welch's $t$ test. We used the Mann–Whitney $U$ test to assess statistically significant differences for the moisture class variable. Our null hypothesis was that there is no difference in landscape characteristics between the distributions of fires with warming or cooling impacts. In other words, fires with climate-warming impacts do not occur in significantly drier landscapes than fires with climate-cooling impacts. Sample sizes differ among landscape categories because the data originate from a synthesis field database[38], where individual field campaigns collected different subsets of variables, leading to uneven variable availability between categories and across warming and cooling fires.

We used 30 m resolution annual land-cover maps from 2000 to 2014 over the ABoVE core domain to identify pre-fire land cover for all fires between 2001 and 2015[89]. We used the simplified 10-class land-cover maps, in which evergreen forests are defined as a combination of evergreen forest and woodlands, while the deciduous class includes deciduous and mixed forest stands. The land-cover classification was resampled using spatial averaging of all 30 m pixels within 500 m pixels to derive fractional cover of each land-cover class within every 500 m pixel. We applied a threshold of 90% fractional cover to contrast almost pure evergreen and deciduous forests. This resulted in 416,281 evergreen needleleaf pixels and 5,233 deciduous broadleaf pixels. We assessed the statistical differences in the distributions of net climate impacts in evergreen and deciduous forests using the Mann–Whitney $U$ test. Our null hypothesis was that there is no difference in the distribution of net climate impacts between evergreen and deciduous forests. In addition, we report the percentage of pixels within each forest type that show net climate-warming or climate-cooling effects. To understand what drives the differences in net radiative forcing, we analysed the relative contribution of each forcing component. This is expressed as percentage change and defined by how much the mean climate impacts in deciduous forests differ from those in evergreen forests, using evergreen forests as a baseline reference.

In addition, we analysed fire characteristics using long-term fire records and the ABoVE-FED database[41]. Similar to the analysis focused on landscape characteristics, we classified fires as being climate warming or climate cooling. However, this classification was based on the average net climate impact of the entire fire perimeter. We included 11,795 individual fires for this analysis. This aggregation to fire level applies to the variables fire size, total carbon combusted, day of burning and burn depth in Table 1. Furthermore, we used the northern treeline delineation from the Circumpolar Arctic Vegetation Map (CAVM)

to define the treeline in our analysis[37]. To quantify the distance from the treeline for each location, we used the Euclidian distance from each pixel to the nearest treeline boundary.

Some analyses were conducted at pixel scale, while others were performed at fire level, due to differences in data resolution and availability. Analyses based on the field database were limited by the number of field sites and measured parameters. However, when sufficient data existed, we aggregate measurements at the fire level, which then allows us to consider all pixels within each fire for subsequent spatial analyses.

### Uncertainty in radiative forcing estimates

We calculated relative uncertainties in radiative forcing, expressed as a percentage of the value estimates. These uncertainties are presented in the figures and text in W m$^{-2}$ of burned area. We created ten different versions of our radiative forcing framework for greenhouse gases, precursors and aerosols. These model versions were developed by randomly sampling the parameter spaces within one standard deviation of specified emissions factors[66] and GWPs[67,68] (Supplementary Table 1). We assessed the uncertainty in this framework using 10,000 experimentally burned pixels. The carbon combustion values were randomly derived from a normal distribution using the mean value of 3.13 kgC m$^{-2}$ and a standard deviation of 1.20 kgC m$^{-2}$ from the ABoVE-FED database[32]. The uncertainty in our radiative forcing estimates from greenhouse gas and precursor emissions was 36% (relative uncertainty expressed as a percentage), while the uncertainty in radiative forcing from aerosols was 85% (relative uncertainty expressed as a percentage). The substantial uncertainty in radiative forcing estimates for aerosols is due to the propagation of major uncertainties in the emissions factors and GWPs of black and organic carbon. In addition, there is unquantified uncertainty due to the transport of aerosols away from the fire.

We used random forest regressors to predict trajectories of post-fire albedo and vegetation recovery. To estimate uncertainty, we constructed ten distinct bootstrapped models per post-fire month and year for surface albedo and ten bootstrapped models per post-fire year for the vegetation recovery. Each model was trained using 60% of all available training pixels. These models were then used to generate predictions across a set of 10,000 pixels, which were randomly sampled from all burned pixels between 2001 and 2019. The standard deviation was computed to represent the prediction uncertainty while the mean absolute value across all models was used as the measured value. Our uncertainty estimates for radiative forcing due to changes in surface albedo and vegetation recovery were 16% and 8% (relative uncertainties expressed as a percentage). We did not account for the inclusion of unburned islands within the historical fire database perimeters that could introduce pixels from unburned islands into our 70-year chronosequences of post-fire albedo changes and net ecosystem $CO_2$ exchange. Such unburned islands would mostly be present for fires earlier in the time series.

We used a sensitivity analysis to estimate uncertainties related to carbon emissions from fire-induced permafrost thaw. We aimed to address key uncertainties in the total carbon release after thaw and the post-fire changes in active layer thickness. In our model, we used results from lab incubations to derive carbon release curves after thaw (Supplementary Fig. 6b) and applied a correction factor[86] to better resemble field conditions. In addition, we derived post-fire active layer thickness curves from in situ measurements across discontinuous permafrost landscapes in boreal North America, leveraging a unique 50-year post-fire time series[51,87]. Our first-order approach might underestimate active layer thickness changes across the diverse permafrost conditions of our study domain (Supplementary Fig. 6a) due to persistent knowledge gaps, such as active layer thickness dynamics after the fire with different severities, subsidence and permafrost properties[49]. We did not account for subsidence, which would lead to an underestimation of soil carbon newly exposed to thaw. However, our carbon release curves are modelled on the basis of permafrost

soils that were incubated at a constant temperature of 5 °C. In reality, these soils may be at 5 °C or warmer for parts of the summer but will be colder and frozen for much of the year. Hence, we incorporated a seasonality factor based on the number of frost-free days in a given year and a lab-to-field scaling factor based on temperature sensitivity[86]. Thus, our approach may spatially lead to both overestimates and underestimates of carbon emissions. We established six different model scenarios, including a version without the lab-to-field scaling factor of 2.93 (ref. [86]), a version with carbon emissions reduced by 50%, another with emissions increased by 50%, one with a 50% additional thickening of the active layer, and versions that combined changes in carbon emission estimates with additional active layer thickening (Supplementary Fig. 8). We randomly selected 10,000 burned pixels that were underlain by permafrost for this sensitivity analysis. The relative uncertainty in radiative forcing resulting from fire-induced permafrost emissions was 53%.

We did not incorporate uncertainties within the simplified radiative expressions. These expressions were used in our models of greenhouse gases and precursors, vegetation recovery and permafrost to compute radiative forcing of well-mixed greenhouse gases. The radiative expressions are applicable to atmospheric concentrations within specific ranges: 180–2,000 ppm for $CO_2$, 200–525 ppb for $N_2O$ and 340–3,500 ppb for $CH_4$ (ref. [26]). These limits are well within the ranges of atmospheric concentrations projected under the SSP2–4.5 climate scenario. The simplified expressions are often used for quicker calculations or when integrating into larger climate models where computational efficiency is critical. However, these simplified models can introduce errors. Etminan et al.[26] used the Oslo line-by-line model to benchmark the accuracy of these expressions. The Oslo line-by-line model is a highly detailed and precise computational method used to calculate radiative forcing. It involves detailed spectral analysis of how gases absorb and emit radiation at individual wavelengths and is used for precise atmospheric radiative transfer calculations. The absolute error for $CO_2$ was 3.6% (0.15 W m$^{-2}$), for $N_2O$ the error was 0.64% (0.003 W m$^{-2}$), and for $CH_4$ the error was 2.7% (0.016 W m$^{-2}$) (ref. [26]). There is also uncertainty stemming from the radiative kernels used to calculate radiative forcing from changes in albedo, resulting from nonlinearities in the albedo feedback[90]. This uncertainty may evolve as cloud properties undergo changes with climate change, directly influencing how much radiation is scattered, absorbed or emitted within the atmosphere. We did not account for these uncertainties in our framework. We used the latest available radiative kernels[82], and the use of these kernels may explain some of the differences with previous radiative forcing estimates of post-fire changes in albedo across boreal North America that were based on different radiative kernels[20,21,90]. For our post-fire surface albedo and vegetation recovery, the use of historical fire records introduces spatial bias towards reported fires before 1965. The earlier records are dominated by fires reported in the southern boreal region, with a lower winter and spring surface albedo and higher canopy density when compared with fires near the latitudinal treeline[22].

### Data availability

The Arctic–Boreal Vulnerability Experiment fire emissions database can be accessed from https://daac.ornl.gov/cgi-bin/dsviewer.pl?ds_id=2063 (ref. [41]). MODIS-Derived Daily Mean Blue-Sky Albedo can be downloaded from https://daac.ornl.gov/cgi-bin/dsviewer.pl?ds_id=1605 (ref. [77]). Net ecosystem exchange data can be obtained from https://daac.ornl.gov/cgi-bin/dsviewer.pl?ds_id=2377 (ref. [83]). The climate projections used in this study can be accessed from https://adapt-west.databasin.org/pages/adaptwest-climatena/ (ref. [80]). Long-term fire perimeter databases from Alaska (ref. [64]) can be accessed from https://fire.ak.blm.gov/predsvcs/maps.php and those from Canada (refs. [15,63]) can be accessed from https://cwfis.cfs.nrcan.gc.ca/data-mart/download/lfdb. The Global Multi-resolution Terrain Elevation

Data 2010 digital elevation model can be accessed from https://www.usgs.gov/centers/eros/science/usgs-eros-archive-digital-elevation-global-multi-resolution-terrain-elevation. SoilGrids data (ref. 79) can be obtained from https://github.com/ISRICWorldSoil/SoilGrids250m. The permafrost zonation and ruggedness index can be downloaded from https://microsite.geo.uzh.ch/cryodata/pf_global/ (ref. 65). The high-spatial-resolution soil organic carbon data (ref. 84) can be downloaded from https://bolin.su.se/data/palmtag-2022-spatial-1. Data for Figs. 1–3 and Extended Data Figs. 2–6 are available via Zenodo at https://doi.org/10.5281/zenodo.18327524 (ref. 91).

## Code availability
All analyses were conducted using Python software 3.11.10. The code deemed central for the analyses in this paper is available via https://zenodo.org/records/15719840 (ref. 92).

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

## Acknowledgements
We thank the European Research Council for funding support through a Consolidator Grant under the European Union's Horizon 2020 research and innovation programme (grant agreement number 101000987) (S.V.). B.M.R. acknowledges support from the National Aeronautics and Space Administration (NASA) Arctic–Boreal Vulnerability Experiment (ABoVE; NNX15AU56), the National Science Foundation Arctic System Sciences (grant number 2116864) and funding catalysed by the Audacious Project (Permafrost Pathways). S.J.G. acknowledges support from NASA ABoVE grants 80NSSC19M0113 and 80NSSC22K1247. Q.Z. acknowledges support from the National Natural Science Foundation of China (NSFC 42477392) and the NSFC Excellent Young Scientists Fund Program (Overseas). M.C.M. acknowledges support from US National Science Foundation grants OPP-2116862 and DEB-2224776 and the USDA Forest Service, PNW Research Station. J.T.R. acknowledges support from NASA's Modeling, Analysis, and Prediction (80NSSC21K1362), ABoVE (80NSSC23K0140) and Earth Information System–Fire research programmes and from the US National Science Foundation (RISE-2425932). We thank J. Watts and M. Farina for advising on the use of a net ecosystem exchange dataset. We also thank L. Heffernan for his advice on the permafrost radiative forcing framework.

## Author contributions

M.J.v.G., S.V. and B.M.R. designed the research; M.J.v.G. performed the analysis with input from S.V., B.M.R, S.C., S.P., M.M., A-M.V. and Q.Z.; B.M.R. and S.V. acquired the funding; S.V. administered and supervised this project. The manuscript was drafted by M.J.v.G., S.V. and B.M.R.; S.J.G., M.C.M., J.T.R., N.S., M.R.T. and G.R.v.d.W. provided comments and contributed to manuscript revisions. All authors participated in reviewing and editing the manuscript.

## Competing interests
The authors declare no competing interests.

## Additional information
**Extended data** is available for this paper at https://doi.org/10.1038/s41561-026-01940-3.

**Correspondence and requests for materials** should be addressed to
Max J. van Gerrevink.

**Peer review information** *Nature Geoscience* thanks Víctor
Fernández-García, Simone Stuenzi and Elizabeth Webb for their
contribution to the peer review of this work. Primary Handling Editor:
Carolina Ortiz Guerrero and Aliénor Lavergne, in collaboration
with the *Nature Geoscience* team. Peer reviewer reports are
available.

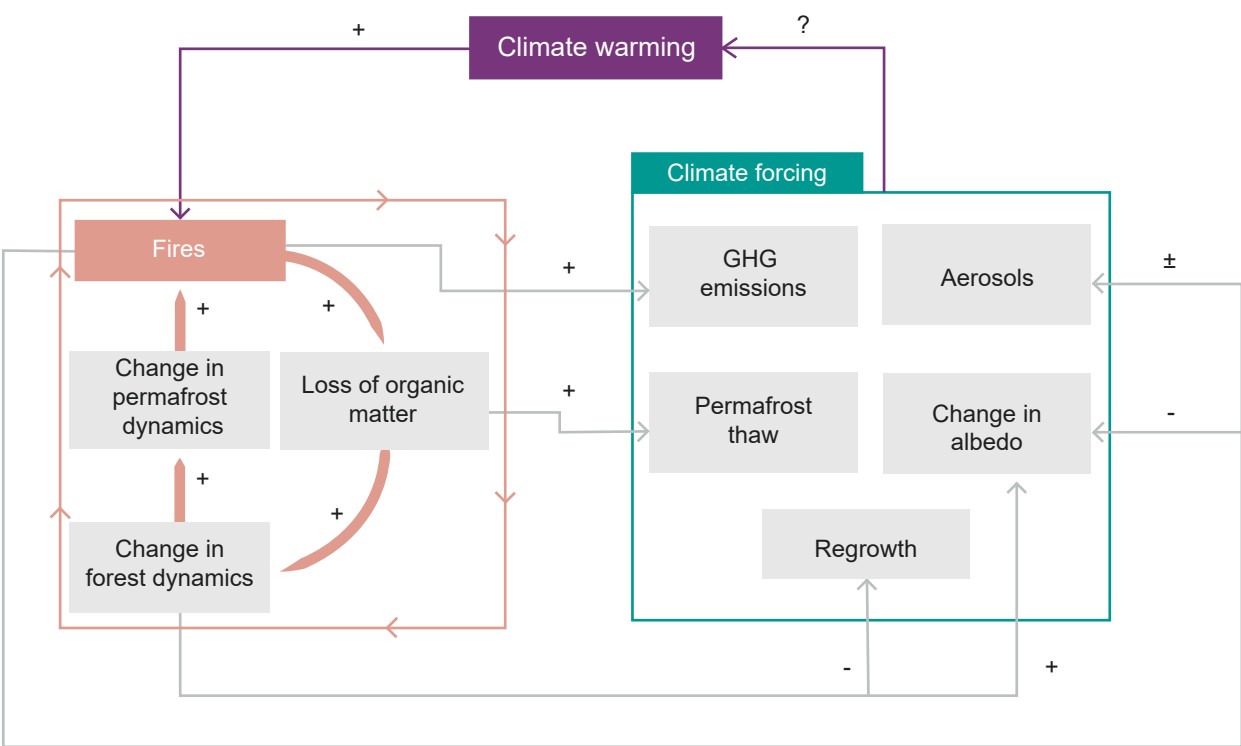

Fire-vegetation-permafrost feedback

Carbon-climate feedback

**Extended Data Fig. 1 | Schematic illustration of how climate warming influences the fire-vegetation-permafrost feedback and the carbon-climate feedback loops in boreal forest ecosystems.** The gray arrows highlight the pathways in which the fire-vegetation-permafrost feedback loop initiates positive and negative climate forcing. Fires have positive and negative feedbacks by directly emitting greenhouse gases (GHG) and aerosols during combustion.

The loss of canopy and organic matter can cause permafrost degradation and associated greenhouse gas emissions. After fires, forests recover and carbon sequestration occurs. Post-fire surface albedo changes in boreal forests primarily result in a negative radiative forcing, because of persistent snow exposure in recently burned areas during winter and spring.

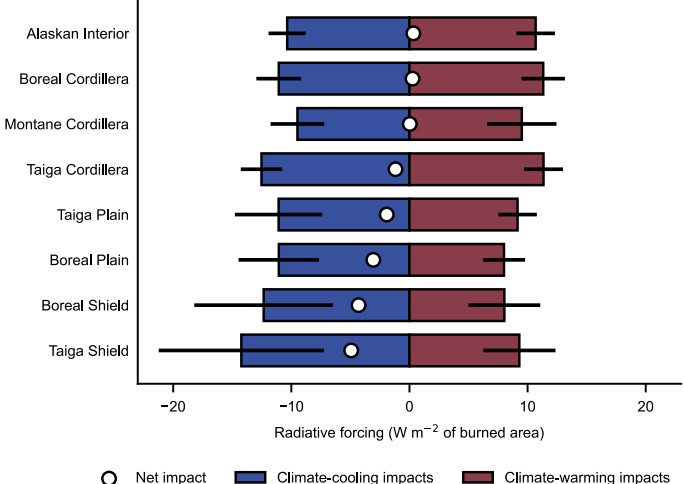

**Extended Data Fig. 2 | Contribution plot of climate-warming (red bars), climate-cooling (blue bars) and net impacts (white circles) from fires between 2001 and 2019 across ecoregions over a 70-year period.** Climate-warming impacts show the combined impact of direct greenhouse gas and precursor emissions and permafrost greenhouse gas emissions. Climate-cooling impacts show the combined impact from post-fire changes in surface albedo, aerosol emissions and post-fire vegetation recovery. Error bars show one standard deviation of spatial variability around the mean climate-warming or climate-cooling impacts within each ecoregion.

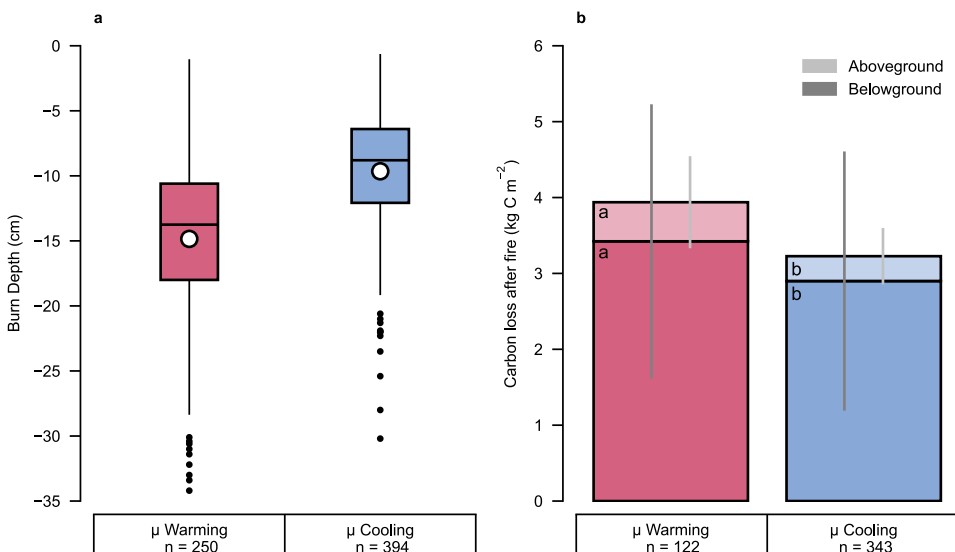

**Extended Data Fig. 3 | Burn depth and carbon combustion from above- and below ground carbon pools as function of net climate impacts.**
**a**, Climate-warming fires (red; n = 250) burned deeper into the soil organic layer than climate-cooling fires (blue; n = 394) (p < 0.001; two-sided Welch's t-tests). Horizontal lines represent the median, the white circle represents the mean and upper and lower limits of the boxes show the 25th and 75th percentiles. Whiskers extend up to 1.5 times the interquartile range. Outliers beyond this range are highlighted with black dots. **b**, Average ecosystem carbon pool losses after fire across climate-warming (red; n = 122) and climate-cooling fires (blue; n = 343).

Darker colors represent the belowground carbon pools, the lighter colors represent the aboveground carbon pools. The error bars represent the standard deviation around the mean of carbon pool losses (gray; belowground, and silver; aboveground). Letters represent significant differences (p < 0.001) between the two groups within the above- and belowground pools. Statistical significance was assessed using two-sided Welch's t-tests. The data in the figure is from the Arctic-Boreal Vulnerability Experiment (ABoVE) synthesis field dataset of combustion measurements[38]. Pixels were classified according to their mean net climate radiative forcing into warming and cooling pixels.

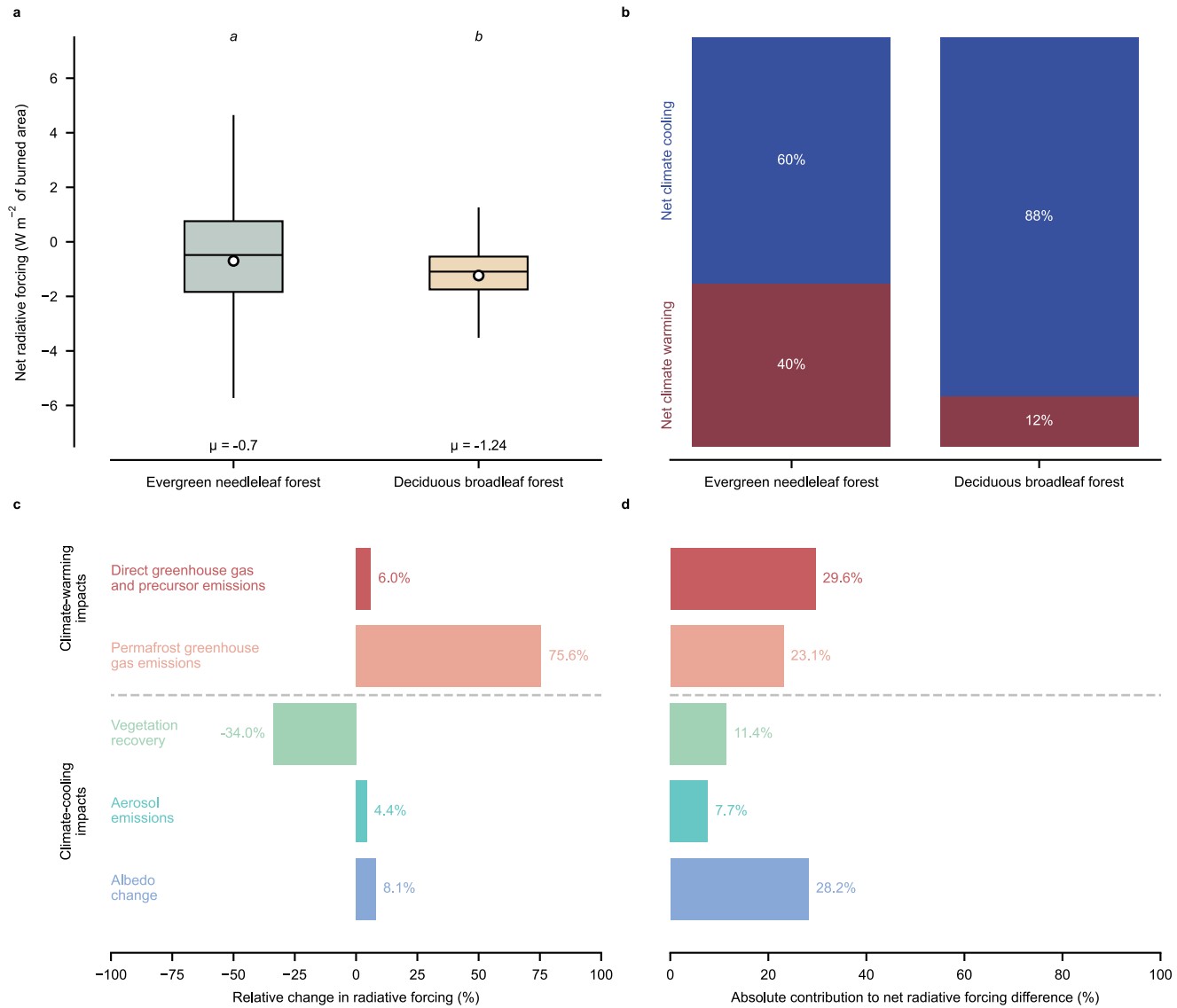

**Extended Data Fig. 4 | Comparison of climate impacts between evergreen needleleaf and deciduous broadleaf forests. a**, Pixel-based comparison of the net radiative forcing between evergreen needleleaf forests (green) and deciduous broadleaf forests (yellow) (p < 0.001). Horizontal lines represent the median, the white circle represents the mean and upper and lower limits of the boxes show the 25th and 75th percentiles. Whiskers extend up to 1.5 times the interquartile range. Letters represent significant differences (p < 0.001) between evergreen needleleaf and deciduous broadleaf forests. Statistical significance was assessed using two-sided Welch's t-tests. **b**, Stacked bar chart of net climate impact categories. The red bar represents the percentage of pixels within each forest type that display net climate-warming impacts, and the blue bars represent the percentage of pixels within each forest type that show net climate-cooling

impacts. **c**, The relative change in evergreen needleleaf forest values compared to deciduous broadleaf forest values, expressed as a proportion of the evergreen needleleaf forest mean. Positive relative changes mean that the average climate impact is greater in evergreen needleleaf forests than in deciduous broadleaf forests. Negative relative changes mean that the average climate impact is smaller in evergreen needleleaf forests than in deciduous broadleaf forests. **d**. Absolute contribution to the net radiative forcing difference between evergreen needleleaf and deciduous broadleaf forests. Larger values indicate a greater contribution to the overall difference between evergreen needleleaf and deciduous broadleaf forests. This figure is based n = 416,281 evergreen needleleaf forest pixels and 5,233 deciduous broadleaf forests pixels (see Methods).

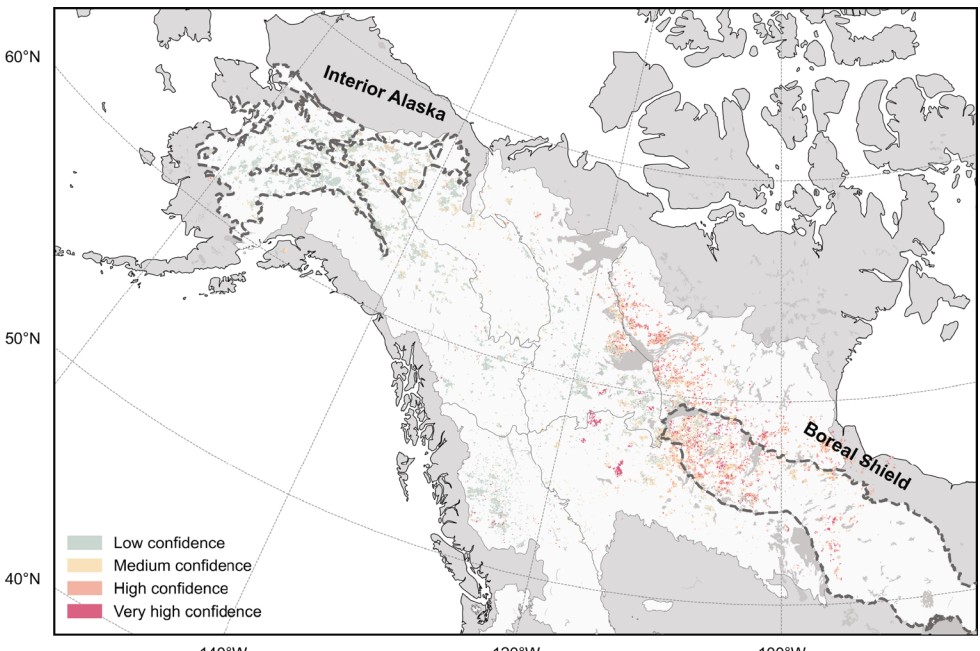

**Extended Data Fig. 5 | Spatially-explicit representation of confidence in net climate impacts.** The map illustrates the confidence in the sign of net climate effects, showing how certain we are that a given area contributes to either climate warming (positive radiative forcing) or climate cooling (negative radiative forcing). Low confidence corresponds to a probability range of 50-66%, moderate confidence to 66-80%, high confidence to 80-90%, and very high confidence to probabilities exceeding 90%. The ecoregion boundary for Interior Alaska and the Boreal Shield are delineated by the dashed gray lines. The ecoregion boundary for Interior Alaska and the Boreal Shield are delineated by the dashed gray lines. Ecoregion data from the US Environmental Protection Agency (https://www.epa.gov/eco-research/ecoregions-north-america). Basemap from Natural Earth (https://www.naturalearthdata.com).

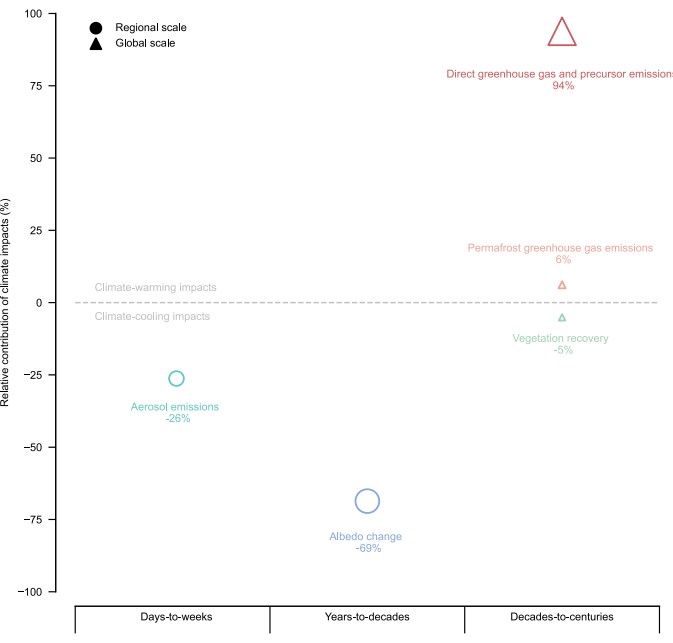

**Extended Data Fig. 6 | Schematic illustration of the temporal and spatial footprints of different climate impacts from boreal fires and their relative contributions in percentage to the total warming and cooling impacts over time.** Positive values (above the dashed line) indicate contributions to warming, while negative values (below the dashed line) indicate contributions to cooling in percentages. The x-axis represents the temporal footprint, ranging from short-term effects (days-to-weeks) to long-term effects (decades-to-centuries). The different symbols denote the spatial footprint of each impact, distinguishing between regional (circles) and global-scale (triangles).

**Extended Data Table 1 | The net radiative forcing and its contributing factors over the 70-year period in Alaska, western Canada, and the combined western boreal North American (Alaska and western Canada) region**

| Variable | | Alaska | Western Canada | Western boreal North America |
|---|---|---|---|---|
| Net radiative forcing with permafrost | | 0.35 (±4.66) W m$^{-2}$ | -2.88 (±4.17) W m$^{-2}$ | -1.99 (±4.29) W m$^{-2}$ |
| Net radiative forcing without permafrost | | -1.05 (±4.60) W m$^{-2}$ | -3.25 (±4.17) W m$^{-2}$ | -2.64 (±4.28) W m$^{-2}$ |
| Climate-warming impacts | Direct greenhouse gas and precursor emissions | 9.51 (±3.43) W m$^{-2}$ | 8.78 (±3.17) W m$^{-2}$ | 8.98 (±3.24) W m$^{-2}$ |
| | Permafrost greenhouse gas emissions | 1.40 (±0.74) W m$^{-2}$ | 0.37 (±0.19) W m$^{-2}$ | 0.65 (±0.34) W m$^{-2}$ |
| Climate-cooling impacts | Albedo change | -6.62 (±1.06) W m$^{-2}$ | -8.71 (±1.40) W m$^{-2}$ | -8.14 (±1.31) W m$^{-2}$ |
| | Aerosol emissions | -3.38 (±2.88) W m$^{-2}$ | -2.73 (±2.32) W m$^{-2}$ | -2.90 (±2.47) W m$^{-2}$ |
| | Vegetation recovery | -0.56 (±0.04) W m$^{-2}$ | -0.59 (±0.05) W m$^{-2}$ | -0.58 (±0.05) W m$^{-2}$ |

Forcing estimates are given in W m$^{-2}$ of burned area. Uncertainty is presented as one standard deviation of the mean prediction uncertainty.

