## [Peer Review File · Nature Geoscience]

Climate impacts from North American boreal forest fires

Corresponding Author: Mr Max van Gerrevink

Version 0:

Reviewer comments:

Reviewer #1

(Remarks to the Author)

This is my third review of Climate impacts from North American boreal forest fires by van Gerrevink and colleagues. The manuscript is an important contribution to quantifying the radiative effect of North American boreal forest fires, particularly in that it models radiative forcing over a large spatial area and a large number of fires. In their response to my earlier comments, the authors stated that “explicitly accounting for fire-induced permafrost emissions is one of the important breakthroughs of the paper.” While I think that the data is not available to make such broad proclamations, I do think the authors have provided enough reasoning for their methods and quantification of the permafrost emissions factor that their estimate is scientifically justified.

The author’s explanation for their new approach to quantifying the permafrost emissions component is well reasoned. My only feedback on this topic is that a substantial portion of their justification in the response to reviewers was not also included in the manuscript text. I think other readers would benefit from this discussion and I urge the authors to include it in the supplementary material. Specifically, I am referring to the paragraph beginning with “The NEE model from Virkkala et al” and “Permafrost modeling framework estimates..”

In my first and second review, I suggested a figure showing how the contribution of each RF component changes over time. This is a standard way of visualizing RF from fires (See for example: O’Halloran et al., 2012; Randerson et al., 2006) and is instructive in understanding how components differ across timescales and ecosystems. As I mentioned previously, this also allows the reader to see how the choice of a 70-year time frame affects the results and could facilitate conceptual projections of how changes to fire intervals might affect the cumulative RF effect of fires. It also provides an important window into the data, which can be hard to interpret when presented as an aggregated value. I do not find the authors’ argument that “overlapping signals from hundreds of individual fires would likely obscure meaningful patterns rather than clarify them” to be convincing, since that is essentially what science is: extracting meaning from messy signals. This figure could be broken up by ecozone or vegetation type, if the authors think that would help clarify the processes.

Overall, the manuscript would benefit from improved clarity of the Methods. While the details are well explained for each RF component, I think I am missing a big-picture explanation. For example, it is not until the supplementary methods that the authors clarify that they are considering the 70-year lifecycle of fires that occurred in the 2001-2019 interval, rather than the effect of historical fires during 2001-2019. (From the abstract: “estimate climate impacts from boreal fires in Alaska and western Canada between 2001 and 2019” could be interpreted either way.) It is never explicitly stated where the 2001-2019 fire perimeters come from, and how the study fires were delineated is not even mentioned in the main methods (the main methods do mention the AK Large Fire Database and the Canadian National Fire Database, but in the context of the space-for-time analysis of historical fires; are we to infer the same dataset is used for delineating the 2001-2019 fires that form the basis of this study?). As far as I can tell, the total number of fires, which seems important (!) is only mentioned in a figure caption, rather than in the Methods. It is not clear to me why some analyses are done on the fire scale while others are done at the pixel scale. I don’t understand why the sample size in Table 1 varies between variables (e.g., total sample size for elevation is 420, stand density is 513, etc. but there are 11,795 total fires). To remedy these issues, I suggest adding 1-2 paragraphs at the beginning of the Methods that provides a conceptual overview of the study methods. What is the high-level overview of the study design without all the details?

80 what is the ‘burned area’ unit here? A pixel? A fire?

119 This sentence seems out of place, since the paragraph is on proximity to tree line and this sentence is on the importance of PF thaw-induced GHG emissions. I suggest this sentence be integrated into the following paragraph.

141 This seems like an important point: that the difference between warming fires vs. cooling fires is due to how much they C they combust. Is this the main difference or just one of many? Can the authors add a sentence or two describing why fires

would combust more or less C (fire weather, antecedent conditions, etc.). This seems like an important point for understanding the future climate effects of fires in the NAM boreal region.

147 This second half of the paragraph could benefit from a sentence or two explaining the mechanisms behind the differences between fire-induced RF in different forest types. For example, is the reason more evergreen forest fires are warming because of albedo recovery? Why is the contribution of PF emissions different in each PFT? Is this because of PF conditions that underlay the different forest types?

165 I don't think dominated is the right word here. Surpassed?

372 reference needed

460 The main text indicates that treeline analysis was done on a per-fire basis, but these methods suggest the distance was calculated on a per-pixel basis. Please clarify.

553 Per Nature guidelines, information about access to primary datasets (generated during the study) and referenced datasets (datasets analyzed in the study) must be provided. The authors used many more datasets in their analysis than are listed here. For example, fire perimeters, NEE dataset, etc.

676 If you're only looking at fires 2001-2019, why is a Landsat-based database not appropriate? Is this because you need older fires to quantify albedo trajectories? I think being a bit more explicit here would help the reader understand.

Fig 1: The blue and orange boundaries are hard to decipher because they are within the same color palette of the RF gradient. I suggest changing the boundary colors.

Fig 2: Are the values presented at the top of the figure the slope and p-value? That should be noted in the figure caption. Please provide p-values for all regression lines, as suggested by the Nature submission guidelines.

Fig 3: The interpretation of white circles is listed twice.

Table 1: How many fires in each category (warming vs cooling)? Why do some variables have descriptors as to the data origination (e.g., 'Burn depth (ABOVE-FED)') whereas others do not (e.g., elevation)? Either is fine, but consistency across variables would be nice. Additionally, I am puzzled by the use of "-" to distinguish the number of warming fires from the number of cooling fires, since typically the dash is typically used to convey subtraction. Perhaps W (C) or W/C would be better than (W) - (C).

Extended Table Figure 1: I suggest clarifying the title. Something along the lines of: "The net radiative forcing of fires over the 70-year period averaged across geographic regions..."

Extended Figure 4: Same comment as fig 1 regarding the color of the boundaries.

Extended Figure 6: What is the unit of comparison here? Fires or pixels? What is n? The y axis on part C is grey and difficult to read. I suggest making the labels black. For the description of part d, please add what positive values indicate, as you did in the description of part c.

(Remarks on code availability)

Reviewer #2

(Remarks to the Author)

The authors have once again substantially improved the quality of the manuscript, effectively addressed my remaining comments and further strengthened the discussion of the studies' limitations. The inclusion of more explicit statements on the potential for ecosystem shifts and their interaction with changes in hydrothermal permafrost conditions and active layer dynamics is valuable. I appreciate the careful revisions and the expanded discussion on differences in the timing and magnitude of vegetation recovery-caused carbon sequestration, as well as the addition of the lab-to-field scaling factor which limits the potential overestimation of emissions from permafrost thaw.

One potential enhancement could be to include a concise summary sentence in the discussion section, and perhaps also in the abstract, clarifying which key processes are represented (e.g., surface albedo, greenhouse gas and aerosol emissions from fire, permafrost extent) and which are not (e.g., vegetation succession or shifts, thermokarst).

I have only a two minor remarks:

- The addition of blue and orange delineations in Figure 1 and throughout the manuscript helps distinguish the regions; however, using these colors which also correspond to the main color scale in Fig. 1 creates some visual confusion and alters the overall impression of the maps. It may be clearer to differentiate the regions using background shading (e.g., darker and lighter grey tones) or another neutral visual cue that does not interfere with the data colors. This would likely

improve clarity without adding unnecessary visual complexity.

- I appreciate the discussion on the potential impacts of abrupt thaw events. This section could be further strengthened by constraining it with the spatial distribution of excess ground ice (e.g., Brown et al., 2002), which would provide at least a rough indication of areas most susceptible to such processes. The statement, "Our estimates of post-fire permafrost emissions are likely conservative because we could not incorporate post-fire subsidence and abrupt thaw events due to data and knowledge limitations," remains very general. The potential underestimation mentioned is limited to thermokarst-prone areas and cannot be used to negate the potential for overestimation of the permafrost thaw component elsewhere. Including even an approximate percentage or spatial extent of regions where abrupt thaw is likely would make the argument more robust and quantitatively grounded. Particularly within the Yedoma domain of eastern Siberia, excess ice and its potential for abrupt thaw have been more extensively studied, including its links to the initiation and duration of surface subsidence and thermokarst formation.

Brown, J., O. Ferrians, J. A. Heginbottom, and E. Melnikov. 2002. Circum-Arctic Map of Permafrost and Ground-Ice Conditions, Version 2. [Indicate subset used]. Boulder, Colorado USA. NASA National Snow and Ice Data Center Distributed Active Archive Center.

(Remarks on code availability)

Reviewer #3

(Remarks to the Author)

In my opinion, this revised version of "Climate impacts from North American boreal forest fires" is publishable in Nature Geoscience after minor revisions. The reasons for this recommendation are below:

(I) The authors have satisfactorily addressed one of the major concerns raised during the review process (the reliability of greenhouse gas emissions from fire-induced permafrost thaw). They have now applied a scaling factor to account for the difference between laboratory and field emission rates, following the results of Natali et al. (2019). Importantly, the authors have also validated their permafrost emission estimates against field measurements, showing a good agreement between modeled and observed fluxes. This substantially strengthens the credibility of their permafrost radiative forcing framework.

(II) Regarding my main concern in the previous revision (the issue of spatial autocorrelation) the authors have acknowledged the distinction between temporal and spatial dependencies and provided some justification for the potentially limited influence of spatial autocorrelation on their analyses. However, I recommend that the authors succinctly clarify in the manuscript why the model performance in Potter et al. (who explicitly accounted for spatial autocorrelation) should be expected to resemble the models presented here, succinctly emphasizing their shared characteristics (e.g., same response variable, similar geographic region, comparable predictor sets). This may help readers understand better the rationale of the sentence.

(III) Finally, the authors should explicitly indicate the error metric used in L60 ($\pm 0.62 \text{ kg C m}^{-2}$ and the other one), as well as in the abstract (L26).

I hope my comments had been helpful and constructive.

(Remarks on code availability)

NOTE: This file includes the line-by-line responses to the reviewers. This document refer to modification made throughout the main text and Supplementary Information. The lines correspond to the lines presented in the modification_line_by_line_comments.pdf file.

Reviewer #1 (Remarks to the Author):

This is my third review of Climate impacts from North American boreal forest fires by van Gerrevink and colleagues. The manuscript is an important contribution to quantifying the radiative effect of North American boreal forest fires, particularly in that it models radiative forcing over a large spatial area and a large number of fires. In their response to my earlier comments, the authors stated that “explicitly accounting for fire-induced permafrost emissions is one of the important breakthroughs of the paper.” While I think that the data is not available to make such broad proclamations, I do think the authors have provided enough reasoning for their methods and quantification of the permafrost emissions factor that their estimate is scientifically justified.

The author’s explanation for their new approach to quantifying the permafrost emissions component is well reasoned. My only feedback on this topic is that a substantial portion of their justification in the response to reviewers was not also included in the manuscript text. I think other readers would benefit from this discussion and I urge the authors to include it in the supplementary material. Specifically, I am referring to the paragraph beginning with “The NEE model from Virkkala et al” and “Permafrost modeling framework estimates..”

Thank you for your very helpful comments and dedication in reviewing our manuscript. We agree that the readers would benefit from this explanatory text. Therefore, we have now integrated this text from the previous response letter in the supplementary material.

Lines 1133-1158: “The NEE model from Virkkala et al.^{58,85}, included only 7% of the training data from burned forests on permafrost terrain, of which more than 75% are in isolated (0-10% coverage) and discontinuous (10-50% coverage) permafrost terrain. Hence fire-related NEE impacts tied to permafrost thaw are underrepresented and fluxes from active layer thickening responding to fire are likely missed. The underestimation bias is critical because permafrost soils store vast amounts of carbon, that can release substantial amounts of CO₂ and CH₄. Although the existing model effectively captures vegetation and upper-soil processes through broader environmental predictors⁵⁸, the dataset has likely limited sensitivity to permafrost thaw emissions. The dataset estimates post-fire NEE dynamics reasonably well after fire⁵⁸, nonetheless underestimates net CO₂ emissions in fire disturbed sites, particularly during the initial post-fire years (up to 7–10 years). This limitation contributes to documented underestimations, where carbon sources after fire are underestimated by up to 75 g C m⁻² month⁻¹, with an average of 5.2 ± 23.1 g C m⁻² month⁻¹ ⁵⁸. This is why we introduced a more direct integration of fire-induced permafrost emissions to provide a more comprehensive understanding of post-fire carbon budget (See Methods and supplementary section 2.5).

Our permafrost modeling framework estimates that 83 percent of all fire-induced permafrost emissions occur within the first two decades after the fire, totaling on average roughly 2.6 ± 1.2 kg C m⁻² as CO₂ over this period. This translates to counterbalancing a persistent net CO₂ bias of approximately 10.9 ± 5.1 g C m⁻² month⁻¹ in the existing NEE models over a 20-year timeframe. This estimate is approximately twice the magnitude of underestimations reported by Virkkala et al.⁵⁸,

yet these values are likely conservative. We could not incorporate post-fire subsidence and abrupt thaw events due to data and knowledge limitations. Turetsky et al.⁵⁰ estimated that abrupt thaw contributes a radiative forcing comparable to that from gradual thaw, effectively doubling the total permafrost-related climate impact when both processes would be considered.”

In my first and second review, I suggested a figure showing how the contribution of each RF component changes over time. This is a standard way of visualizing RF from fires (See for example: O’Halloran et al., 2012; Randerson et al., 2006) and is instructive in understanding how components differ across timescales and ecosystems. As I mentioned previously, this also allows the reader to see how the choice of a 70-year time frame affects the results and could facilitate conceptual projections of how changes to fire intervals might affect the cumulative RF effect of fires. It also provides an important window into the data, which can be hard to interpret when presented as an aggregated value. I do not find the authors’ argument that “overlapping signals from hundreds of individual fires would likely obscure meaningful patterns rather than clarify them” to be convincing, since that is essentially what science is: extracting meaning from messy signals. This figure could be broken up by ecozone or vegetation type, if the authors think that would help clarify the processes.

We thank the reviewer for the helpful suggestion. We agree that visualizing the temporal evolution of radiative forcing contributions is a standard and informative approach, and it can indeed clarify how different components operate over time and under varying fire regimes. Importantly, our study provides pixel-based estimates of radiative forcing, highlighting the substantial heterogeneity in post-fire climate impacts within individual fires. Individual fires vary widely in size, location, and timing, and aggregating hundreds of fire-specific trajectories at the domain scale risks creating an overly complex and potentially misleading representation.

To balance clarity and informativeness, we have included Figure S9 following the last round of revisions, which illustrates the temporal evolution of radiative forcing components for the 2009 Minto Flats South fire. This example not only provides a clear and interpretable depiction of radiative forcing dynamics over time but also demonstrates how the choice of a 70-year time frame affects the results and facilitates conceptual projections of how changes in fire intervals may influence cumulative radiative forcing effects.

Overall, the manuscript would benefit from improved clarity of the Methods. While the details are well explained for each RF component, I think I am missing a big-picture explanation. For example, it is not until the supplementary methods that the authors clarify that they are considering the 70-year lifecycle of fires that occurred in the 2001-2019 interval, rather than the effect of historical fires during 2001-2019. (From the abstract: “estimate climate impacts from boreal fires in Alaska and western Canada between 2001 and 2019” could be interpreted either way.) It is never explicitly stated where the 2001-2019 fire perimeters come from, and how the study fires were delineated is not even mentioned in the main methods (the main methods do mention the AK Large Fire Database and the Canadian National Fire Database, but in the context of the space-for-time analysis of historical fires; are we to infer the same dataset is used for delineating the 2001-2019 fires that form the basis of this study?). As far as I can tell, the total number of fires, which seems important (!) is only mentioned in a figure caption, rather than in the Methods. It is not clear to me why some analyses are done on the fire scale while others are done at the pixel scale. I don’t understand why the sample size in Table 1 varies between variables (e.g., total sample size for elevation is 420, stand density is 513, etc. but there are 11,795 total fires). To remedy these issues, I suggest adding 1-2 paragraphs at the beginning of the Methods that provides a conceptual overview of the study methods. What is the high-level overview of the study design without all the details?

Thank you for your valuable suggestions. We have improved and clarified the Methods section. First we included an overview of our methods to help explain the bigger picture as suggested. In the overview paragraph, we restate our definition of climate radiative forcing and we explicitly mention that we derived the burned area from the Arctic-boreal Vulnerability Experiment fire database. This paragraph is formulated on lines 283-295.

Lines 283-295: “We estimated the net radiative forcing impacts from North American boreal forest fires between 2001 and 2019 relative to a no-fire baseline. Our estimates are expressed in units of $W m^{-2}$ of burned area and represent the integrated climate impact over a 70-year post-fire period under the shared socioeconomic pathway SSP2-4.5³⁰. We used burned area data from the Arctic-boreal Vulnerability Experiment fire emissions database (ABOVE-FED)³³ and quantified the net climate radiative forcing impacts at 500 m resolution. In this work, net radiative forcing represents the balance of multiple fire-driven climate forcing components, integrating greenhouse gas and aerosol emissions from combustion, changes in surface albedo, post-fire vegetation recovery and greenhouse gas emissions associated with fire-induced permafrost thaw. We measured these metrics in relation to the treeline to capture spatial and ecological variation in fire-driven radiative forcing across boreal forests. Finally, we reported landscape and forest characteristics at the pixel level and fire-specific attributes at the fire-level to examine their influence on the net climate radiative forcing.”

We have revised parts of the abstract and Online Methods. It is now explicitly noted that our results are over a 70-year period and are representative for ongoing climate change under shared socioeconomic pathway SSP2-4.5. This is stated on lines: 26-30, 66-68 and 284-286.

Lines 26-30: “Here, we estimate climate impacts from boreal fires in Alaska and western Canada between 2001 and 2019 using integrated net radiative forcing metrics combining greenhouse gas and aerosol emissions from combustion, vegetation recovery, greenhouse gas emissions from fire-induced permafrost thaw, and changes in surface albedo over a 70-year period.”

Lines 66-68: “Our radiative forcing estimates, – in units of $W m^{-2}$ of burned area, – capture climate impacts over a 70-year post-fire period (supplementary section 1), using the “Middle-of-the-road” shared socioeconomic pathway SSP2-4.5³⁰.”

Lines 284-286: “Our estimates are expressed in units of $W m^{-2}$ of burned area and represent the integrated climate impact over a 70-year post-fire period under the shared socioeconomic pathway SSP2-4.5³⁰.”

We clarified why some analyses were done at the pixel level and others at the fire level. Additionally, we explain why the sample size differs between landscape categories and specifically state how many fires are included in this assessment on lines 576-577. We stated this motivation on lines 550-553 and 583-587.

Lines 576-577: “We included 11,795 individual fires for this analysis.”

Lines 550-553: “Sample sizes differ among landscape categories because the data originates from a synthesis field database³⁹, where individual field campaigns

collected different subsets of variables, leading to uneven variable availability between categories and across warming and cooling fires.”

Lines 583-587: “Some analyses were conducted at pixel scale, while others were performed at fire level, due to differences in data resolution and availability. Analyses based on the field database were limited by the number of field sites and measured parameters. However, when sufficient data existed, we aggregate measurements at the fire level, which then allows us to consider all pixels within each fire for subsequent spatial analyses.”

line 80 what is the ‘burned area’ unit here? A pixel? A fire?

We have included have included the units to clarify what is meant by combustion per unit of burned area on lines 80-83.

Lines 80-83: “In northwestern and western boreal North America, combustion per unit of burned area (kg C m^{-2}) is comparatively higher due to deeper burning into organic soils, leading to more direct greenhouse gas emissions³¹.”

line 119 This sentence seems out of place, since the paragraph is on proximity to tree line and this sentence is on the importance of PF thaw-induced GHG emissions. I suggest this sentence be integrated into the following paragraph.

Thank you for this suggestion. We have revised this section of the text and moved it subsequently to the following paragraph to better align with the storyline. This section is moved to lines 152-160.

Lines 152-160: “In large parts of the Interior Alaska, Montane Cordillera and Boreal Cordillera ecoregions, higher fuel consumption and thus emissions of greenhouse gases and precursors offsets the albedo-driven cooling, leading to climate-warming fires (Extended Figure 2). Notably, the absence of permafrost in the Montane Cordillera and the southern part of the Boreal Cordillera further distinguishes their fire dynamics and associated climate impacts from those in other ecoregions. In Interior Alaska, our findings highlight a substantial role for greenhouse gas emissions from fire-induced permafrost thaw. Without accounting for post-fire greenhouse gas emissions from permafrost thaw, Alaskan fires would result in an average climate-cooling effect of $-1.05 \pm 4.60 \text{ W m}^{-2}$ (Extended Table 1).”

line 141 This seems like an important point: that the difference between warming fires vs. cooling fires is due to how much they C they combust. Is this the main difference or just one of many? Can the authors add a sentence or two describing why fires would combust more or less C (fire weather, antecedent conditions, etc.). This seems like an important point for understanding the future climate effects of fires in the NAM boreal region.

Thank you for pointing this out. We added a sentence explaining that differences in carbon combustion largely reflect landscape and hydrological conditions, which influence fire intensity, depth of burn, and total carbon release per unit area.

Lines 179-180: “The difference in combustion reflects landscape and hydrological characteristics that promote deeper and more severe burning, and larger carbon release per unit area.”

line 147 This second half of the paragraph could benefit from a sentence or two explaining the mechanisms behind the differences between fire-induced RF in different forest types. For example, is the reason more evergreen forest fires are warming because of albedo recovery? Why is the contribution of PF emissions different in each PFT? Is this because of PF conditions that underlay the different forest types?

Thank you for this suggestion. We have incorporated an explanatory sentence on this topic including a reference to Mack et al. (2021), which describes this.

Lines 193-196: "The pattern reflects vegetation traits, as evergreen needleleaf forests accumulate thicker organic soils and exhibit longer recovery rates, whereas deciduous broadleaf forests promote faster carbon turnover and shallower permafrost, limiting post-fire warming impacts¹⁰."

line 165 I don't think dominated is the right word here. Surpassed?

We have changed the wording in this statement on this line to exceeded:

Lines 207-210: "Similar to previous work done at individual field sites^{13,22}, the cooling influence of post-fire surface albedo changes in such ecosystems exceeded the warming effects from greenhouse gases emitted at the time of fire and from permafrost thaw."

line 372 reference needed

Thank you for pointing this out. We have included the reference to Virkkala et al. (2025) on lines 470-472.

Lines 470-472: "This documented bias is approximately half of the reported permafrost carbon emissions from active layer thickening after fire⁵⁸."

58. Virkkala, A.-M. et al. Wildfires offset the increasing but spatially heterogeneous Arctic–boreal CO₂ uptake. Nat Clim Chang <https://doi.org/10.1038/s41558-024-02234-5> (2025) doi:10.1038/s41558-024-02234-5.

line 460 The main text indicates that treeline analysis was done on a per-fire basis, but these methods suggest the distance was calculated on a per-pixel basis. Please clarify.

We have clarified the main text to align better with the methodology. This clarification is provided on lines 131-132 by stating that it is per unit area burned.

Lines 131-132: "We further examined the climate impacts of fires in Alaska and Canada per unit area burned in relation to the treeline³⁸."

Line 553 Per Nature guidelines, information about access to primary datasets (generated during the study) and referenced datasets (datasets analyzed in the study) must be provided. The authors used many more datasets in their analysis than are listed here. For example, fire perimeters, NEE dataset, etc.

Thank you for pointing this out. We have expanded to include all datasets central to the analysis.

Lines 693-708: “The Arctic-Boreal Vulnerability Experiment fire emission database can be accessed from https://daac.ornl.gov/cgi-bin/dsviewer.pl?ds_id=2063 (ref.⁴²). MODIS-Derived Daily Mean Blue-Sky Albedo downloaded from https://daac.ornl.gov/cgi-bin/dsviewer.pl?ds_id=1605 (ref.⁷⁶). Net ecosystem exchange data can be obtained from https://daac.ornl.gov/cgi-bin/dsviewer.pl?ds_id=2377 (ref.⁸³). The climate projections used in this study can be accessed from <https://adaptwest.databasin.org/pages/adaptwest-climatena/> (ref.⁸⁰). Long-term fire perimeter databases from Alaska (ref.⁷³) can be accessed from <https://fire.ak.blm.gov/predsvcs/maps.php> and Canada (ref.^{16,74}) from <https://cwfis.cfs.nrcan.gc.ca/datamart/download/lfdb>. The Global Multi-resolution Terrain Elevation Data 2010 digital elevation model can be accessed from <https://www.usgs.gov/centers/eros/science/usgs-eros-archive-digital-elevation-global-multi-resolution-terrain-elevation>. SoilGrids data (ref.⁷⁹) can be obtained from <https://github.com/ISRICWorldSoil/SoilGrids250m>. The permafrost zonation and ruggedness index can be downloaded from https://microsite.geo.uzh.ch/cryodata/pf_global/ (ref.⁷⁸). The high spatial resolution soil organic carbon data (ref.⁸⁴) can be downloaded from <https://bolin.su.se/data/palmtag-2022-spatial-1>. Data for Figures 1-3 and Extended Data Figures 2-6 are available via Zenodo at: <https://doi.org/10.5281/zenodo.18327524> (ref.⁹²)”

Line 676 If you're only looking at fires 2001-2019, why is a Landsat-based database not appropriate? Is this because you need older fires to quantify albedo trajectories? I think being a bit more explicit here would help the reader understand.

We used MODIS blue-sky albedo rather than Landsat-derived estimates because blue-sky albedo accounts for both direct and diffuse solar radiation, capturing realistic surface reflectance under typical sky conditions. In contrast, Landsat-based albedo is derived from broadband measurements under mostly clear-sky conditions and does not explicitly account for diffuse light. This mismatch can lead to biases when estimating the radiative impact of surface albedo changes, particularly in post-fire landscapes where albedo recovery is gradual and influenced by diffuse radiation. We have incorporated a motivation for this within the Online Methods on lines 353-357:

Lines 353-357: “We used the MODIS albedo product over Landsat-derived estimates because MODIS provides consistent daily observations with a spectral resolution well-suited for robustly quantifying post-fire surface albedo. In contrast, Landsat’s sparse temporal coverage and spectral bands limit its ability to capture gradual, landscape-scale albedo changes relevant for climate radiative forcing calculations.”

Fig 1: The blue and orange boundaries are hard to decipher because they are within the same color palette of the RF gradient. I suggest changing the boundary colors.

Thank you for your suggestion. We have revised our maps in figure 1 to highlight the boundaries of Interior Alaska and the Boreal Shield better. This has been done using a dotted gray line and adding a label for both ecoregions. The figure caption has been revised to reflect these changes. Additionally, we revisited every map in our entire manuscript.

Lines 717-726: “Figure 1 | Cumulative mean climate radiative forcing from fires between 2001 and 2019 across Alaska and western Canada over a 70-year period. a, Net radiative forcing map. b, Contribution of net radiative forcing (black circles), direct greenhouse gas and precursor emissions (red squares), permafrost greenhouse gas emissions (orange upside-down triangles), change in surface albedo (blue triangles), aerosol emissions (cyan hexagons), and vegetation recovery (green diamonds) for Interior Alaska and Boreal Shield fires. The error bars indicate the uncertainty of each agent with one standard deviation based on the uncertainty assessment. c, Net radiative forcing over some selected fires in Interior Alaska. d, Net radiative forcing over some selected fires in central Canada. The ecoregion boundary for Interior Alaska and the Boreal Shield are delineated by the dashed gray lines. Ecoregion data were obtained from the United States Environmental Protection Agency (EPA) (<https://www.epa.gov/eco-research/ecoregions-north-america>, last accessed 21 January 2026). Basemap in panels a, c and d are made with Natural Earth.”

Fig 2: Are the values presented at the top of the figure the slope and p-value? That should be noted in the figure caption. Please provide p-values for all regression lines, as suggested by the Nature submission guidelines.

Thank you for your helpful comment. We have updated Figure 2, including its caption. P-values are now provided for each regression line, including the ones that are not

significant. The caption was revised to mention the slope as suggested by the reviewer. The figure caption reads now as follows:

Line 728-735: “Figure 2 | Fire radiative forcing as a function of distance to the latitudinal treeline for (a) Alaska and (b) western Canada. Net radiative forcing (black circles), direct greenhouse gas and precursor emissions (red squares), permafrost greenhouse gas emissions (orange upside-down triangles), vegetation recovery (green diamonds), aerosol emissions (cyan hexagons), and change in surface albedo (blue triangles) trends in relation to distance to treeline are shown as mean values within 50 km intervals. The shading represents the 95% confidence interval around the regression line. Individual trends are shown with slopes and p-values. Statistical significance was determined by a two-sided t-test of the regression coefficient ($p < 0.05$, $p < 0.01$, and $p < 0.001$).”

Fig 3: The interpretation of white circles is listed twice.

Thank you for pointing this out. We have corrected the figure caption. The figure caption reads now as follows:

Lines 737-746: “Figure 3 | Radiative forcing from permafrost greenhouse gas emissions induced by fire across landscapes with different permafrost extents. a, Radiative forcing from permafrost greenhouse gas emissions under different permafrost landscapes from all fires across the study domain between 2001 and 2019. b, The net radiative forcing for different permafrost landscapes. Fires from the Interior Alaska are shown in green, Boreal Shield fires in yellow, and domain-wide estimate across western boreal North America in orange. Permafrost extent was defined using the permafrost zonation index from Gruber⁸⁰. Burned pixels per permafrost landscape; Interior Alaska: isolated ($n = 6,897$), sporadic ($n = 1,303,778$), discontinuous ($n = 868,043$), continuous ($n = 80,944$). Boreal Shield: isolated ($n = 336,406$), sporadic ($n = 1,492,235$), discontinuous ($n = 19,607$), continuous ($n = 0$). Western Boreal North America: isolated ($n = 1,127,980$), sporadic ($n = 5,018,978$), discontinuous ($n = 3,631,818$), continuous ($n = 371,920$). Note, the Boreal Shield

ecoregion does not have landscapes with continuous permafrost. Horizontal lines represent the median, the white circle represents the mean and upper and lower limits of the boxes show the 25th and 75th percentiles. Whiskers extend up to 1.5 times the interquartile range.”

Table 1: How many fires in each category (warming vs cooling)? Why do some variables have descriptors as to the data origination (e.g., ‘Burn depth (ABOVE-FED)’) whereas others do not (e.g., elevation)? Either is fine, but consistency across variables would be nice. Additionally, I am puzzled by the use of “-“ to distinguish the number of warming fires from the number of cooling fires, since typically the dash is typically used to convey subtraction. Perhaps W (C) or W/C would be better than (W) – (C).

We have adjusted the way we distinguish between warming and cooling fires. To improve clarity, we have adjusted the column name to: “Sample size (W) | (C)”. The letters W and C are explained in the Table caption. Moreover, we have clarified the data sources in the table footer. We now explicitly mention which method and source were used to derive the results.

Table 1 | Landscape and fire characteristics of climate-warming and climate-cooling fires (s.d. = standard deviation, W = sample size climate-warming fires, C = sample size climate-cooling fires, number of ‘All fires’ = 11,795).

Variable	$\mu_{\text{climate-warming}}$ fires (\pm S.d.)	$\mu_{\text{climate-cooling}}$ fires (\pm S.d.)	P-value	Sample size (W) (C)
Net radiative forcing	0.84 (\pm 0.61) W m ⁻²	-3.37 (\pm 4.06) W m ⁻²		All fires
Moisture class*	Subxeric	Subhygric	0.05	(138) (476)
Slope	6.20 (\pm 7.19) °	2.23 (\pm 5.01) °	<0.001	(134) (269)
Elevation	553 (\pm 236) m	322 (\pm 126) m	<0.001	(149) (271)
Stand density	0.67 (\pm 0.63) stems m ⁻²	0.69 (\pm 0.71) stems m ⁻²	0.81	(99) (414)
Stand age	103 (\pm 51) years	96 (\pm 49) years	0.21	(134) (344)
Fraction black spruce (Pre-fire)	0.85 (\pm 0.28)	0.71 (\pm 0.36)	<0.001	(151) (402)
Total C combustion (ABOVE-FED)	3.49 (\pm 0.38) kg C m ⁻²	2.81 (\pm 0.76) kg C m ⁻²	<0.001	All fires
Burn depth (ABOVE-FED)	11.6 (\pm 1.7) cm	9.6 (\pm 2.3) cm	<0.001	All fires
Fire size (long-term databases)	6.62 (\pm 21.6) × 1,000 ha	7.01 (\pm 27.3) × 1,000 ha	0.51	All fires

Day of burning (long-term databases)	199 (± 48) day of year	187 (± 37) day of year	<0.001	All fires
--	------------------------------	------------------------------	------------------	-----------

* * The variable moisture class is a categorical dataset. We present the mode of climate-warming and climate-cooling fires in this case. The P-value of the moisture class was assessed with a two-sided Mann-Whitney U test. All other P-values were evaluated using two-sided Welch's t-test. The variables moisture class, slope, elevation, stand density, stand age, and fraction black spruce (pre-fire) are all pixel-based and are derived from the Arctic-boreal Vulnerability Experiment synthesis field dataset of combustion measurements³⁹. The variables total C combusted, burn depth, fire size and day of burning are all fire perimeter-based and are derived from the Arctic-boreal Vulnerability Experiment Fire Emission Database⁴² and long-term governmental fire databases from Alaska and Canada^{16,75,76}. References for data sources are given in the Methods. Bold font in the P-value column refers to P-values smaller than 0.05. Exact P-values are provided whenever possible."

Extended Table Figure 1: I suggest clarifying the title. Something along the lines of: "The net radiative forcing of fires over the 70-year period averaged across geographic regions..."

We have adopted the proposed changes in the table caption of Extended Table 1.

Lines 793-797: "Extended Table 1 | The net radiative forcing and its contributing factors over the 70-year period in Alaska, western Canada, and the combined western boreal North American (Alaska and western Canada) region. Forcing estimates are given in $W\ m^{-2}$ of burned area. Uncertainty is presented as one standard deviation of prediction uncertainty."

Extended Figure 4: Same comment as fig 1 regarding the color of the boundaries.

Extended Figure 4 has been revised. We have updated every map throughout the whole manuscript to match and to highlight the boundaries of Interior Alaska and the Boreal Shield better. We have chosen to highlight the ecoregions with gray dashed lines and a text label. The figure in question has been relabeled to be Extended Data Figure 5.

Lines 802-808: “Extended Figure 5 | Spatially explicit representation of confidence in net climate impacts. The map illustrates the confidence in the sign of net climate effects, showing how certain we are that a given area contributes to either climate warming (positive radiative forcing) or climate cooling (negative radiative forcing). Low confidence corresponds to a probability range of 50-66%, moderate confidence to 66-80%, high confidence to 80-90%, and very high confidence to probabilities exceeding 90%. The ecoregion boundary for Interior Alaska and the Boreal Shield are delineated by the dashed gray lines. The ecoregion boundary for Interior Alaska and the Boreal Shield are delineated by the dashed gray lines. The ecoregion boundary for Interior Alaska and the Boreal Shield are delineated by the dashed gray lines. Ecoregion data were obtained from the United States Environmental Protection Agency (EPA) (<https://www.epa.gov/eco-research/ecoregions-north-america>, last accessed 21 January 2026). Basemap in figure is made with Natural Earth.”

Extended Figure 6: What is the unit of comparison here? Fires or pixels? What is n? The y axis on part C is grey and difficult to read. I suggest making the labels black. For the description of part d, please add what positive values indicate, as you did in the description of part c.

Thank you for pointing this out. We are making a pixel-based comparison here. This is now explicitly stated in the Figure caption. We have included the number of pixels within each plant functional type in the figure caption and in the methods section. The y-axis labels in panel C are now black. We also included a description for panel D. In short, larger values show a greater contribution to the absolute difference between evergreen needleleaf forests and deciduous broadleaf forest. This figure has been relabeled to be Extended Data Figure 4.

Lines 826-845: “Extended figure 4 | a Comparison of climate impacts between evergreen needleleaf and deciduous broadleaf forests. a, Pixel-based comparison of the net radiative forcing between evergreen needleleaf forests (green) and deciduous broadleaf forests (yellow) ($p < 0.001$). Horizontal lines represent the median, the white circle represents the mean and upper and lower limits of the boxes show the 25th and 75th percentiles. Whiskers extend up to 1.5 times the interquartile range. Letters represent significant differences ($p < 0.001$) between evergreen needleleaf and deciduous broadleaf forests. Statistical significance was assessed using two-sided Welch’s t-tests. b, Stacked bar chart of net climate impact categories. The red bar represents the percentage of pixels within each forest type that display net climate-warming impacts, and the blue bars represent the percentage of pixels within each forest type that show net climate-cooling impacts. c, The relative change in evergreen needleleaf forest values compared to deciduous broadleaf forest values, expressed as a proportion of the evergreen needleleaf forest mean. Positive relative changes mean that the average climate impact is greater in evergreen needleleaf forests than in deciduous broadleaf forests. Negative relative changes mean that the average climate impact is smaller in evergreen needleleaf forests than in deciduous broadleaf forests. d. Absolute contribution to the net radiative forcing difference

between evergreen needleleaf and deciduous broadleaf forests. Larger values indicate a greater contribution to the overall difference between evergreen needleleaf and deciduous broadleaf forests. This figure is based on $n = 416,281$ evergreen needleleaf forest pixels and 5,233 deciduous broadleaf forest pixels (see Methods)”

Lines 562-563: “This resulted in 416,281 evergreen needleleaf pixels and 5,233 deciduous broadleaf pixels.”

Reviewer #2 (Remarks to the Author):

The authors have once again substantially improved the quality of the manuscript, effectively addressing my remaining comments and further strengthened the discussion of the studies’ limitations. The inclusion of more explicit statements on the potential for ecosystem shifts and their interaction with changes in hydrothermal permafrost conditions and active layer dynamics is valuable. I appreciate the careful revisions and the expanded discussion on differences in the timing and magnitude of vegetation recovery-caused carbon sequestration, as well as the addition of the lab-to-field scaling factor which limits the potential overestimation of emissions from permafrost thaw. One potential enhancement could be to include a concise summary sentence in the discussion section, and perhaps also in the abstract, clarifying which key processes are represented (e.g., surface albedo, greenhouse gas and aerosol emissions from fire, permafrost extent) and which are not (e.g., vegetation succession or shifts, thermokarst).

We sincerely thank the reviewer for these thoughtful comments. We have included a concise statement on which processes are represented in the abstract. This is in the revised manuscript on lines 26-30. We chose not to include an additional statement in the discussion section as the second paragraph thoroughly discusses the represented components. The omitted components have been mentioned in the third paragraph of the discussion. This paragraph provides the justification for not including them. Moreover, this would push the word limit for the main text significantly over the set threshold.

Lines 26-30: “Here, we estimate climate impacts from boreal fires in Alaska and western Canada between 2001 and 2019 using integrated net radiative forcing metrics combining greenhouse gas and aerosol emissions from combustion, vegetation recovery, greenhouse gas emissions from fire-induced permafrost thaw, and changes in surface albedo over a 70-year period.”

I have only a two minor remarks:

The addition of blue and orange delineations in Figure 1 and throughout the manuscript helps distinguish the regions; however, using these colors which also correspond to the main color scale in Fig. 1 creates some visual confusion and alters the overall impression of the maps. It may be clearer to differentiate the regions using background shading (e.g., darker and lighter grey tones) or another neutral visual cue that does not interfere with the data colors. This would likely improve clarity without adding unnecessary visual complexity.

Thank you for this suggestion. We have addressed this comment to align with the comments by Reviewer 1. All maps have been updated accordingly to highlight the boundaries of Interior Alaska and the Boreal Shield better. We have delineated the ecoregions with a dotted gray line and have added a label for both ecoregions. Please see our response to Reviewer 1 above.

I appreciate the discussion on the potential impacts of abrupt thaw events. This section could be further strengthened by constraining it with the spatial distribution of excess ground ice (e.g., Brown et al., 2002), which would provide at least a rough indication of areas most susceptible to such processes. The statement, “Our estimates of post-fire permafrost emissions are likely conservative because we could not incorporate post-fire subsidence and abrupt thaw events due to data and knowledge limitations,” remains very general. The potential underestimation mentioned is limited to thermokarst-prone areas and cannot be used to negate the potential for overestimation of the permafrost thaw component elsewhere. Including even an approximate percentage or spatial extent of regions where abrupt thaw is likely would make the argument more robust and quantitatively grounded. Particularly within the Yedoma domain of eastern Siberia, excess ice and its potential for abrupt thaw have been more extensively studied, including its links to the initiation and duration of surface subsidence and thermokarst formation.

Brown, J., O. Ferrians, J. A. Heginbottom, and E. Melnikov. 2002. Circum-Arctic Map of Permafrost and Ground-Ice Conditions, Version 2. [Indicate subset used]. Boulder, Colorado USA. NASA National Snow and Ice Data Center Distributed Active Archive Center.

We thank the reviewer for this constructive suggestion. We have revised the discussion to more explicitly constrain the potential influence of abrupt thaw processes using the spatial distribution of ground ice. Specifically, we now reference the circumpolar ground ice classification of Brown et al. (2002) to emphasize that abrupt thaw and thermokarst development are largely confined to ice-rich permafrost regions.

To address the concern that our original statement was overly general, we have clarified that any potential underestimation related to the omission of post-fire subsidence and abrupt thaw processes is limited to ice-rich landscapes. We have also included a first-order, literature-based estimate of the spatial extent of ice-rich permafrost across the boreal forests using data from Nitzbon et al. (2024). According to Nitzbon et al. (2024), high-ice-content landscapes cover approximately a third of the boreal biome. This provides a quantitative and spatially grounded context for the areas where abrupt thaw may influence post-fire permafrost emissions.

Lines 238-244: “Our estimates of post-fire permafrost emissions do not incorporate post-fire subsidence and abrupt thaw events due to data and knowledge limitations, which may lead to conservative estimates for ice-rich permafrost regions. Abrupt thaw events such as thermokarst are primarily confined to ice-rich permafrost landscapes, which represent roughly a third of the pan-boreal region^{48,49}. Abrupt thaw events are hotspots for carbon emissions and can be triggered by single season events such as fires⁵⁰⁻⁵².”

48. Nitzbon, J. et al. No respite from permafrost-thaw impacts in the absence of a global tipping point. *Nat Clim Chang* 14, 573–585 (2024).
49. Brown, J., Ferrians Jr, O. J., Heginbottom, J. A. & Melnikov, E. S. Circum-arctic map of permafrost and ground ice conditions. (2001).
50. Turetsky, M. R. et al. Carbon release through abrupt permafrost thaw. *Nat Geosci* 13, 138–143 (2020).
51. Natali, S. M. et al. Permafrost carbon feedbacks threaten global climate goals. *Proc Natl Acad Sci U S A* 118, (2021).

52. Gibson, C. M. *et al.* Wildfire as a major driver of recent permafrost thaw in boreal peatlands. *Nat Commun* 9, (2018).

Reviewer #3 (Remarks to the Author):

In my opinion, this revised version of “Climate impacts from North American boreal forest fires” is publishable in Nature Geoscience after minor revisions. The reasons for this recommendation are below:

The authors have satisfactorily addressed one of the major concerns raised during the review process (the reliability of greenhouse gas emissions from fire-induced permafrost thaw). They have now applied a scaling factor to account for the difference between laboratory and field emission rates, following the results of Natali *et al.* (2019). Importantly, the authors have also validated their permafrost emission estimates against field measurements, showing a good agreement between modeled and observed fluxes. This substantially strengthens the credibility of their permafrost radiative forcing framework.

We sincerely thank the reviewer for these thoughtful comments. We are pleased to hear that the revised manuscript is now substantially stronger and more credible.

Regarding my main concern in the previous revision (the issue of spatial autocorrelation) the authors have acknowledged the distinction between temporal and spatial dependencies and provided some justification for the potentially limited influence of spatial autocorrelation on their analyses. However, I recommend that the authors succinctly clarify in the manuscript why the model performance in Potter *et al.* (who explicitly accounted for spatial autocorrelation) should be expected to resemble the models presented here, succinctly emphasizing their shared characteristics (e.g., same response variable, similar geographic region, comparable predictor sets). This may help readers understand better the rationale of the sentence.

*Thank you for this suggestion. We have emphasized the shared characteristics of the model in the manuscript with the model presented in Potter *et al.* (2020). We stated that model performance is expected to resemble that of Potter *et al.* because both studies use the same response variable (MODIS blue-sky albedo), cover similar boreal regions in North America, and employ a comparable set of predictor variables. We have included this on lines 396-398.*

Lines 396-398: “This similarity is expected given that we used the same target variable, have an overlapping spatial domain, and relied on comparable features describing bioclimatic, vegetation, and soils characteristics.”

Finally, the authors should explicitly indicate the error metric used in L60 (± 0.62 kg C m⁻² and the other one), as well as in the abstract (L26).

I hope my comments had been helpful and constructive.

Thank you for pointing this out. We have explicitly indicated the error metrics on lines 30-33 and 75-78.

Lines 30-33: “We find that fires across Alaska contributed, on average, to net climate warming (0.35 ± 4.66 W m⁻² of burned area; one standard deviation), while fires across Canada contributed to net cooling (-2.88 ± 4.17 W m⁻² of burned area; one standard deviation).”

Lines 75-78: “However, the total fuel consumption estimates from these individual fire events ($1.76 \pm 0.62 \text{ kg C m}^{-2}$ and $1.73 \pm 0.37 \text{ kg C m}^{-2}$; uncertainties expressed as one standard deviation) were much lower than the average combustion from recent data synthesis and modeling estimates across central and western boreal North America of over 3 kg C m^{-2} ^{20,33,34}.”

Summary of major changes

In the revised version of the manuscript, we have included several major changes to improve the manuscript's methodology and the clarity of our findings.

- We adapted the permafrost radiative forcing framework to incorporate a lab-to-field scaling factor of 2.93, improving the accuracy of our estimates by better representing greenhouse gas emissions from fire-induced permafrost thaw under natural field conditions.
- We used published post-fire soil respiration measurements from the field to validate our permafrost emissions framework.
- We clarified our motivation and reasoning for modeling permafrost emissions and net ecosystem exchange (NEE) of CO₂ as separate components, and we have clarified limitations and uncertainties that may stem from these methodological choices.

The latest code is available at: <https://zenodo.org/records/15719840>.

Referee #1 (Remarks to the Author):

In the revised version, the authors responded to my concerns regarding the permafrost thaw component of their analysis in two ways. First, the authors changed the product they use to measure post-fire vegetation recovery to a newer product modeled on observations. Second, the authors altered their calculation of the contribution of permafrost thaw so that carbon emissions from thawing soils are now only calculated during the non-frozen season rather than year-round. However, the revised manuscript does not adequately resolve these issues.

The updated manuscript relies on an NEE product derived from eddy covariance and chamber measurements (i.e., observational data). NEE, by definition, includes both vegetation and soil (i.e., permafrost) carbon fluxes. The authors are therefore using a product that is observing/modeling NEE and the component fluxes, which includes soil carbon emissions from permafrost thaw. The model used to create the NEE dataset, as stated by the authors and demonstrated by Virkkala et al., (2025), performs reasonably well at sites experiencing recent fire (albeit lower than at sites without disturbance) (see extended data fig 6 from Virkkala et al.). Why then, do the authors additionally add a permafrost thaw component? This would seem to be (1) double counting soil respiration and (2) saying that the models, which apparently perform reasonably well, are missing a massive carbon flux.

Even if I have missed the justification for double counting soil respiration, I still do not think the permafrost thaw component, as estimated in the manuscript, is scientifically justified. As I mentioned in my first review, field conditions are not the same as a controlled laboratory experiment. According to the Natali et al., paper cited by the authors, the temperature sensitivity of lab incubations was nearly 3 times larger than the temperature sensitivity measured in the field. Similarly, Knoblauch et al. (2021) found that short-term incubations estimated fluxes that were 3-4 times larger than in-situ measurements. Simply restricting the analysis period to the number of frost-free days, as the authors have done, does not mitigate this issue. At the very least, the authors need to compare their estimates to post-fire soil respiration measurements from the field to demonstrate their assumptions hold. Given that the NEE estimates compare 'reasonably well' to post-fire measurements without the added modeled permafrost thaw component, I suspect that the assumptions would not hold up.

As the authors themselves demonstrate, this inclusion of C emissions from permafrost thaw has a huge impact on the overall results and is responsible for shifting the impact of fires from cooling to warming in Alaska. Since the choice to use laboratory experiments to estimate emissions from in-situ permafrost thaw lacks justification and there is evidence indicating the estimate is significantly inflated, the results of this paper are not robust.

Thank you for your very helpful comments and dedication in reviewing our manuscript. We agree that controlled lab conditions may not be representative of field conditions. We have now adapted the framework to include a lab-to-field scaling factor of 2.93 based on Natali et al. (2019). Natali et al. (2019) calculated the temperature sensitivity (Q10) of CO₂ emissions based on in situ measurements and laboratory incubations and found notable differences. The Q10 value derived from in situ CO₂ fluxes is 2.9 (95% CI = (2.1, 4.2)) per 10°C soil temperature increase. Conversely, the Q10 value obtained from laboratory incubations is 8.5 (95% CI = (5.0, 14.5)) for CO₂ release from low-temperature laboratory incubations. Comparing these values, the Q10 from laboratory incubations is approximately a factor of 2.93 (8.5/2.9 = 2.93) higher than that from in situ measurements. Implementing the lab-to-field scaling factor in our revised manuscripts results in radiative forcing estimates from fire-induced permafrost emissions that are a factor 2.93 lower than in our previous manuscript.

In addition, we have used post-fire soil respiration measurements from the field to validate our permafrost emissions active layer thickening framework. For doing so, we used chamber measurements on thawing permafrost soils responding to fire from Estop-Aragonés et al. (2018a) and Estop-Aragonés et al. (2018b). Both studies used radiocarbon (¹⁴C) dating to distinguish between respiration from modern plant material and aged permafrost carbon. Based on the results from Estop-Aragonés et al. (2018a), in situ measurements six years post-fire indicate aged soil respiration rates of 0.41 ± 0.20 g CO₂-C m⁻² d⁻¹. For the same site and post-fire interval, we estimate a comparable value of 0.50 ± 0.24 g CO₂-C m⁻² d⁻¹ after applying the scaling factor of 2.93 (Figure S7a). In Figure S7b, we present aged soil respiration rates from Estop-Aragonés et al. (2018b), measured in situ nine years post-fire at 0.21 ± 0.06 g CO₂-C m⁻² d⁻¹. We estimate comparable values of 0.23 ± 0.11 g CO₂-C m⁻² d⁻¹ with the applied scaling factor of 2.93. This supports the use of the scaling factor of 2.93, offering a more accurate estimation of fire-induced permafrost emissions under natural conditions.

References

Estop-Aragonés, C. *et al.* Limited release of previously-frozen C and increased new peat formation after thaw in permafrost peatlands. *Soil Biol Biochem* **118**, 115–129 (2018a).

Estop-Aragonés, C. *et al.* Respiration of aged soil carbon during fall in permafrost peatlands enhanced by active layer deepening following wildfire but limited following thermokarst. *Environmental Research Letters* **13**, (2018b).

Figure S7 | Validation of modeled soil respiration fluxes from post-fire permafrost thaw against in situ data derived from radiocarbon dated chamber flux measurements. a. Comparison of modeled and in situ permafrost carbon fluxes six years post-fire in $\text{g CO}_2\text{-C m}^{-2} \text{ d}^{-1}$ based on Estop-Aragonés et al. (2018a)⁸⁴ b. Comparison of modeled and in situ permafrost carbon fluxes nine years post-fire in $\text{g CO}_2\text{-C m}^{-2} \text{ d}^{-1}$ based on Estop-Aragonés et al. (2018b)⁸⁵. Points represent the in situ $\text{CO}_2\text{-C}$ emissions, with the bars representing the uncertainty as one standard deviation. The teal dotted lines show the model estimates, while the teal shading represents the model uncertainty based on the permafrost sensitivity framework.

Lines 388-397: Natali et al.⁸³ indicated that CO_2 release in controlled laboratory settings is nearly three times more sensitive to temperature changes than what is observed in natural field conditions. Laboratory incubations isolate the effect of temperature on microbial respiration, minimizing the influence of other environmental factors such as soil moisture. Therefore, the controlled conditions in the lab may lead to an overestimation of temperature sensitivity, as they do not capture the buffering effects of other drivers that regulate CO_2 emissions from permafrost soils. Hence, to account for the higher temperature sensitivity observed in controlled laboratory settings compared to in situ field measurements, we applied a scaling factor of 2.93⁸³. We validated the use of this scaling factor by comparing our modelled daily CO_2 fluxes with in situ rates of aged carbon release as CO_2 for a six⁸⁴ and nine⁸⁵ year post-fire environment. The use of the scaling factor provides a more realistic estimate of fire-induced permafrost CO_2 emissions under natural conditions.

Lines 924-929: “However, laboratory incubations provide a controlled environment and isolate the effect of temperature on microbial respiration. The controlled conditions in the lab may therefore lead to an overestimation of temperature sensitivity. To bridge the gap between the lab and field conditions, we included a scaling factor of 2.93⁸⁶ to correct the carbon release curves. This was calculated by comparing the temperature sensitivity CO_2 emissions based on in situ measurements (2.9, 95% CI = [2.1, 4.2]) and laboratory incubations (8.5, 95% CI = [5.0, 14.5])⁸⁶. This scaling

approach provides a more realistic estimate of carbon emissions from post-fire permafrost thaw under natural conditions.”

We also understand the reviewer’s concerns about the potential double counting of these permafrost emissions, as, in theory, these should also be captured by the recovery component based on NEE. We argue, however, that the inclusion of greenhouse gas emissions from permafrost active layer thickening is an important component to address limitations and underestimations of post-fire emissions in the existing NEE model.

The NEE model from Virkkala et al., includes only 7% of the training data from burned forests on permafrost terrain, of which more than 75% are in isolated (0-10% coverage) and discontinuous (10-50% coverage) permafrost terrain. Hence fire-related NEE impacts tied to permafrost thaw are underrepresented and fluxes from active layer thickening responding to fire are likely missed. While the dataset does estimate post-fire NEE dynamics reasonably well after fire (Supplementary Figure 13 from Virkkala et al., 2025), the dataset nonetheless underestimates net CO₂ emissions in fire-disturbed sites, particularly during the initial post-fire years (up to 7–10 years). This limitation contributes to documented underestimations, as shown in Supplementary Figure 14 of Virkkala et al., where carbon sources after fire are underestimated by up to 75 g C m⁻² month⁻¹, with an average of 5.2 ± 23.1 g C m⁻² month⁻¹.

Our permafrost modeling framework estimates that 83 percent of all fire-induced permafrost emissions occur within the first two decades after the fire, totaling on average roughly 2.6 ± 1.2 kg C m⁻² as CO₂ over this period. This translates to counterbalancing a persistent net CO₂ bias of approximately 10.9 ± 5.1 g C m⁻² month⁻¹ in the existing NEE models over a 20-year timeframe. This estimate is approximately twice the magnitude of underestimations reported by Virkkala et al., yet these values likely remain conservative. We could not incorporate post-fire subsidence and abrupt thaw events due to data and knowledge limitations. Turetsky et al. estimate that abrupt thaw contributes a radiative forcing comparable to that from gradual thaw, effectively doubling the total permafrost-related climate impact when both processes would be considered.

In addition, our permafrost modeling framework accounts for permafrost CH₄ emissions, which would remain unaccounted when solely relying on the NEE framework, which is focused on CO₂ fluxes.

We think that explicitly accounting for fire-induced permafrost emissions is one of the important breakthroughs of the paper. We also recognize that our estimates of post-fire permafrost emissions are likely conservative because we could not incorporate post-fire subsidence and abrupt thaw events due to data and knowledge limitations. We realize that there are still limitations to our permafrost framework, which we have discussed in more detail on lines 363-374 of the revised manuscript:

Lines 363-374: “It is important to note that our approach for post-fire vegetation recovery has certain limitations, particularly in capturing the full scope of permafrost carbon emissions. The annual NEE budgets from Virkkala et al., demonstrates a tendency to underestimate net CO₂ emissions in burned areas. One potential reason for the underestimation of CO₂ sources at burned sites is the omission of fire-induced permafrost thaw. While the model may capture some permafrost influences through site data and environmental correlations, it lacks a mechanistic representation of thaw processes and their specific impacts on carbon fluxes. The annual NEE budgets can underestimate carbon release, with biases reaching up to 75 g C m⁻² month⁻¹ at burned sites during cross-validation, with an average of 5.2 ± 23.1 g C m⁻² month⁻¹ ⁵⁶. This documented bias is approximately half of the reported permafrost carbon emissions from active layer thickening after fire. However, our estimates are conservative, as they do not account for post-fire subsidence or abrupt thaw following fire. Carbon emissions from abrupt thaw are estimated to roughly double the radiative impact of permafrost emissions from gradual thaw ⁴⁸.”

Supplementary Figure 14 from Virkkala et al. | The distribution of monthly residuals of NEE. The red vertical line differentiates positive and negative residuals. A negative residual value represents the model overestimating NEE, i.e., underestimating net CO₂ sinks or overestimating net CO₂ sources.

In addition, we have added a paragraph discussing key limitations of the current permafrost framework and outlining potential future directions. We explicitly describe the framework as a first-order estimate, recognizing that it simplifies complex subsurface and ecological processes.

Lines 189-191: “Our estimates of post-fire permafrost emissions are likely conservative because we could not incorporate post-fire subsidence and abrupt thaw events due to data and knowledge limitations.”

Lines 197-199: “Future work should prioritize quantifying post-fire permafrost carbon emissions using in situ measurements to better constrain post-fire carbon releases from permafrost carbon^{53,54}.”

Lines 206-212: “Nevertheless, we chose to model the permafrost emissions and net ecosystem exchange (NEE) of CO₂ separately, as there are indications that the NEE product that we used misses an important source of carbon emissions in post-fire environments⁵⁶. In addition, the NEE model is dependent on data from only a few burned sites in permafrost terrain, and therefore its sensitivity to adequately capture permafrost processes and fluxes may be limited. Future work would benefit from conceptually bringing together carbon dynamics and fluxes from post-fire recovering vegetation and soil respiration from modern plant and aged permafrost carbon.”

Minor comments

Comment 1: As I suggested in my first review, I think a figure showing how each the contributions of each of the RF components changes over time would be extremely useful.

The authors have provided what looks like a conceptual figure of one fire. I was hoping to see the data for the entire region presented in this way.

Thank you for the suggestion. We would like to clarify that the figure presented is not a conceptual diagram, but rather shows actual data for the 2009 Minto Flats South fire. It was selected to illustrate the temporal evolution of radiative forcing components in a clear and interpretable way.

We agree that visualizing how each radiative forcing component evolves over time across the entire domain would be informative. However, due to the substantial spatial and temporal variability in fire characteristics, vegetation types, and post-fire recovery trajectories, such a figure becomes very difficult to interpret when aggregated at the domain scale. The overlapping signals from hundreds of individual fires would likely obscure meaningful patterns rather than clarify them.

We have revised the figure to reflect the major updates made to the permafrost radiative forcing component.

Comment 2: Paragraph beginning with line 83 – this reads like a recitation of a table. What are the mechanisms behind these different values? What ecosystem or fire properties are leading to these differences?

Thank you for this helpful comment. We agree that the original version lacked a clear mechanistic explanation for the observed differences. In response, we have revised the relevant paragraph where needed to include mechanisms such as permafrost extent, vegetation structure and characteristics and snow dynamics.

Lines 85-103: "To examine the influences of fire on climate across ecoregions, we highlight the Boreal Shield and Interior Alaska ecoregions, which show contrasting net climate impacts. On average, fires in the Boreal Shield showed a strong net climate-cooling influence of $-4.23 \pm 3.73 \text{ W m}^{-2}$ (Figure 1d), while fires in Interior Alaska showed a net climate-warming influence of $0.34 \pm 4.56 \text{ W m}^{-2}$ (Figure 1c). The contrasting net climate impacts are embedded within ecosystem and landscape characteristics. The Boreal Shield ecoregion contains little to no permafrost, in contrast to Interior Alaska which lies within the discontinuous permafrost zone (50-90% coverage)³⁶. This biophysical difference results in a greater warming from fire-induced permafrost emissions in Interior Alaska ($1.36 \pm 0.71 \text{ W m}^{-2}$) than in the Boreal Shield ($0.15 \pm 0.08 \text{ W m}^{-2}$; Figure S11). Additionally, post-fire changes in surface albedo led to a stronger climate-cooling impact in the Boreal Shield ($-9.40 \pm 1.51 \text{ W m}^{-2}$) compared to Interior Alaska ($-6.48 \pm 1.04 \text{ W m}^{-2}$), largely driven by differences in forest characteristics and prolonged snow exposure³⁷. In the Boreal

Shield, changes in post-fire surface albedo more than offset the warming caused by greenhouse gas and precursor emissions (i.e., greenhouse gases emitted at the time of the fire; $7.88 \pm 2.84 \text{ W m}^{-2}$) and from fire-induced permafrost thaw ($0.15 \pm 0.08 \text{ W m}^{-2}$). Amplified by the on average cooling responses of aerosol emissions ($2.21 \pm 1.88 \text{ W m}^{-2}$) and post-fire vegetation recovery ($-0.66 \pm 0.05 \text{ W m}^{-2}$), this results in net climate-cooling fires across the Boreal Shield ecoregion. In contrast, in Interior Alaska, climate-cooling from post-fire changes in surface albedo only partially offset the climate-warming effects of the greenhouse gas and precursor emissions ($9.32 \pm 3.36 \text{ W m}^{-2}$). This residual climate warming along with additional permafrost greenhouse gas emissions offsets the climate-cooling from aerosol emissions ($-3.30 \pm 2.81 \text{ W m}^{-2}$) and vegetation recovery (-0.56 ± 0.04) leading to, on average, net climate-warming fires.”

Comment 3: 173 This sentence needs a citation, specifically that GHG emissions are distributed globally.

Thank you for pointing this out. We have added two references to support the statement that greenhouse gas emissions from fires are distributed globally.

Lines 172-175: In contrast, the climate-warming caused by greenhouse gas emissions from fire is long-lived and distributed globally^{45,46}, influencing global temperatures, and impacting policy-relevant metrics such as humanity’s allowable carbon emissions to stay within Paris-aligned global temperature thresholds (Extended Figure 5).

45. Liu, Y., Goodrick, S. & Heilman, W. Wildland fire emissions, carbon, and climate: Wildfire-climate interactions. *For Ecol Manage* **317**, 80–96 (2014).

46. Keppel-Aleks, G. *et al.* Separating the influence of temperature, drought, and fire on interannual variability in atmospheric CO₂. *Global Biogeochem Cycles* **28**, 1295–1310 (2014).

Comment 4: 214 ref needed

Thank you for pointing this out. We have included relevant references to support the statement made in the text.

Lines 219-220: “Losses in surface and deep permafrost soil carbon may be offset by more aboveground carbon sequestration across mixed and deciduous stands^{10,61}.”

10. Mack, M. C. *et al.* Carbon loss from boreal forest wildfires offset by increased dominance of deciduous trees. *Science* (1979) 372, 280–283 (2021).

61. Körner, C. A matter of tree longevity. *Science* (1979) 355, 130–131 (2017).

Referee #2 (Remarks to the Author):

I commend the authors for their thorough and thoughtful responses to all reviewer comments, which have led to substantial improvements in the manuscript. I particularly appreciate the expanded discussion on permafrost-forest cover interactions, which now provides a clearer and more nuanced understanding of these dynamics. The introduction of a seasonality factor for permafrost-thaw-related radiative forcing is appropriate. The added distinction between evergreen needleleaf and deciduous broadleaf stands is a valuable enhancement that strengthens the ecological relevance of the study.

Comment 1: I have a slight remaining concern pertaining to the post-fire vegetation recovery depicted in Figures 1 and 2. Specifically, the recovery trajectories appear nearly identical between the two ecoregions (Interior Alaska and Boreal Shield), which is unexpected given the influence of permafrost conditions on ecosystem recovery following forest fires. While the authors note (lines 196-206) that ecosystem shifts and successional trajectories are not explicitly modeled, but are instead implicitly captured within the post-fire NEE model, the contribution of vegetation recovery to net radiative forcing remains notably small. I would encourage the authors to further clarify the underlying reasons for the apparent lack of differentiation between ecoregions, and to discuss potential limitations or uncertainties associated with this aspect of the modeling framework.

Thank you for this comment. The recovery trajectories appear nearly identical between the two ecoregions. While the recovery trajectories in Figures 1 and 2 may appear visually similar, there are indeed quantitative differences in the radiative forcing associated with vegetation recovery between the two ecoregions. Specifically, we estimate a radiative forcing of $-0.56 \pm 0.04 \text{ W m}^{-2}$ for Interior Alaska (referenced on lines 101-103) and $-0.66 \pm 0.05 \text{ W m}^{-2}$ for the Boreal Shield (referenced on lines 97-99). These differences, though subtle, are meaningful and are more clearly visualized in Supplementary Figure 12.

Figure S12 | Cumulative mean climate radiative forcing from post-fire vegetation recovery from fires between 2001 and 2019 across Alaska and western Canada over a 70-year period.

Lines 97-99: “Amplified by the on average cooling responses of aerosol emissions ($2.21 \pm 1.88 \text{ W m}^{-2}$) and post-fire vegetation recovery ($-0.66 \pm 0.05 \text{ W m}^{-2}$), this results in net climate-cooling fires across the Boreal Shield ecoregion.”

Lines 101-103: “This residual climate-warming along with additional permafrost-driven greenhouse gas emissions offsets the climate-cooling from aerosol emissions ($-3.30 \pm 2.81 \text{ W m}^{-2}$) and vegetation recovery (-0.56 ± 0.04) leading to, on average, net climate-warming fires.”

We acknowledge that our modeling framework does not explicitly account for ecosystem shifts or successional dynamics. Instead, these processes are implicitly captured through the empirical post-fire NEE model, which integrates observed recovery patterns across a range of fire-affected sites. Although vegetation recovery contributes to carbon uptake following fire, its radiative cooling effect is relatively small compared to the warming effect from fire emissions. This contrast arises from the difference in timing and magnitude of carbon exchange. A single pulse emission of CO_2 from fire leads to a disproportionately large radiative forcing because a significant fraction of that carbon remains in the atmosphere for centuries, as described by the impulse response function of Joos et al. (2013) (Figure S1b). In contrast, vegetation recovery sequesters carbon gradually, for example $50 \text{ g CO}_2 \text{ yr}^{-1}$, limiting its ability to offset warming over a 70-year period. Together, these factors explain the modest radiative effect of post-fire recovery, despite differences in ecological context.

Thus, even though the cumulative amount of carbon taken up by recovering vegetation may eventually approach the magnitude of fire emissions, the slower timing and diminishing marginal effect of each gram sequestered limit its effectiveness. Additionally, sequestration solely accounts for CO₂, while direct well-mixed greenhouse gas emissions from fire include a range of other greenhouse gases, such as CH₄ and N₂O. In short, the climate system responds more strongly to the direct release of greenhouse gases from fire than to their gradual sequestration by regrowing vegetation. This asymmetry explains why the radiative impact of vegetation regrowth remains modest. We have included this explanation of temporal asymmetry in the manuscript on lines 704-710 to clarify this mechanism.

Joos, F. *et al.* Carbon dioxide and climate impulse response functions for the computation of greenhouse gas metrics: A multi-model analysis. *Atmos Chem Phys* **13**, 2793–2825 (2013).

Figure S1 | (a) Atmospheric concentration under future climate scenarios in parts per million derived from Meinhausen et al., (2020)³⁰. (b) Impulse response function for different background conditions under different future scenarios (solid lines, reference function for Representative Concentration Pathways (RCP) climate scenarios as simulated with the Bern3D-Lund-Potsdam-Jen model by Joos et al.⁶⁷, dashed-lines represent functions of Shared Socioeconomic Pathways (SSP) climate scenarios).

Lines 704-710: “The modest radiative cooling impacts from vegetation recovery reflects the climate system’s greater sensitivity to the immediate, large pulse of CO₂ emitted by fire. This large impulse persists in the atmosphere for centuries, compared to the gradual carbon uptake by vegetation that occurs over decades. This temporal asymmetry means that the

warming effect of emissions dominates the climate response, limiting the offsetting potential of post-fire carbon sequestration. Over a longer timeframe, the contributions would likely become more comparable, however other well-mixed greenhouse gas emissions may continue to play a significant role.”

On a technical note, the manuscript is now more accessible to a broad readership, while still providing sufficient methodological detail for expert evaluation. The figures are clearer, more visually engaging, include appropriate error bars and captions, and I appreciate the additional visualizations. The introduction and discussion are now more firmly anchored in the relevant literature. The abstract has also been improved, offering a clearer summary of the research and its broader implications. Overall, the manuscript is much improved and is nearing suitability for publication.

We sincerely thank the reviewer for these thoughtful comments. We are pleased to hear that the revised manuscript is now more accessible to a broad readership while retaining sufficient technical detail. We also appreciate the positive comments on the improved clarity and visual quality of the figures, the strengthened literature context in the introduction and discussion, and the enhanced abstract.

Referee #2 (Remarks on code availability):

I have reviewed selected scripts from the provided Python code on a spot-check basis. The code appears to be well-organized, clearly structured, and reasonably well-commented. However, I have not attempted to run any of the code. The README file seems reasonable, all software and package versions required and the data sources are stated.

Thank you for these comments on our code. We have updated the code files of permafrost radiative forcing in this round of revisions. This revision included a scaling factor of 2.93 for permafrost emissions, which is described on lines 388-397 (Online Methods). The latest code is available at: <https://zenodo.org/records/15719840>.

Referee #3 (Remarks to the Author):

The authors have made substantial efforts to improve the quality of the manuscript, addressing many of the previous concerns and enhancing the clarity and structure of the work. I appreciate the revisions and the thoughtful responses provided. I still have some comments for the authors.

Comment 1: Regarding the response to the comment about spatial autocorrelation and the associated changes in the document: splitting the data by month and year primarily addresses temporal autocorrelation, not spatial autocorrelation. Spatial autocorrelation refers to the non-independence of observations in geographic space, and it typically requires assessment to determine its relevance, or/and the use of specific modeling approaches (e.g., spatial error models, inclusion of spatial random effects, or spatially structured covariates). I suggest clarifying this distinction in the manuscript and, if spatial autocorrelation is relevant, applying an appropriate correction or providing a justification for its omission.

Thank you for this thoughtful comment. We agree that our approach of splitting the data by month and year primarily addresses temporal, rather than spatial, autocorrelation. We have clarified the distinction in the revised manuscript. As noted, we did not apply modeling techniques specifically designed to account for spatial autocorrelation, such as spatial error models or spatial random effects. We acknowledge that spatial clustering in environmental conditions introduces some degree of autocorrelation, thereby potentially inflating model performance metrics. However, we note that our post-fire surface albedo models are in close agreement with those of Potter et al., who did explicitly address spatial autocorrelation, suggesting that any inflation is minimal.

Potter et al. mitigates spatial autocorrelation by drawing 10,000 training pixels from a broad spatial domain and weighting them according to the proportion of historically burned pixels within 700 km grid cells. We have revised the manuscript to clarify the distinction between spatial and temporal autocorrelation for the post-fire surface albedo changes on line 305-311.

Lines 305-311: "We do not explicitly account for spatial autocorrelation in our models. As a result, our model performance metrics may be slightly inflated due to the spatial clustering of environmental conditions. Nonetheless, our model performances are very similar to those reported by Potter et al.²¹, who explicitly accounted for spatial autocorrelation. This suggests that any potential inflation caused by spatial autocorrelation is minor and does not compromise the robustness of our models. However, by splitting the data based by month and year, we aim to minimize the influence of spatial dependencies over the time."

Comment 2: Line 34: The sentence sounds a bit awkward. I suggest rephrasing it as “Fire is a major natural disturbance mechanism across boreal North America, triggering both climate-warming (positive) and climate-cooling (negative) effects”.

Thank you for your suggestion. We have rephrased it as suggested.

Lines 34-36: “Fire is a major natural disturbance mechanism across boreal North America, triggering both climate-warming (positive) and climate-cooling (negative) effects^{12–15}.”

Comment 3: Line 73: You indicate the meaning of the uncertainty at Line 107, but not in the previous paragraph. Please ensure consistency throughout the manuscript.

Thank you for this suggestion. We had stated this on line 110 of the revision specifically as this differs from the uncertainty presented throughout the main text. To be more clear and consistent, we have made changes to the manuscript stating on lines 74-76 that the uncertainty represents one standard deviation of prediction uncertainty. This is consistently used throughout the manuscript unless stated otherwise, which is the case on lines 110 and 111.

Lines 74-77: “Over a 70-year period, fires across western boreal North America had a net cooling effect, with a mean radiative forcing of $-0.77 \pm 4.37 \text{ W m}^{-2}$, where the uncertainty represents one standard deviation of the prediction uncertainty (Figure 1; Extended Table 1). This definition of uncertainty is applied throughout, unless stated otherwise.”

Comment 4: Line 133: Fuel load is not a severity indicator, but rather a driver or precursor, along with many others. In the field of burn severity, the word "indicator" typically refers to measures or estimates of fire damage or fuel change after fire occurrence (e.g., dNBR index, Composite Burn Index, tree mortality, or degree of consumption).

Thank you for this comment. We have revised this changing the wording from indicators to drivers.

Lines 133-134: “Pre-fire forest characteristics, such as stand density and species composition, are drivers of fire severity due to their close association with fuel availability^{40,41}.”

Comment 5: Line 156: The discussion is rich in summarizing results and explaining data limitations but is less clear in exposing and organizing mechanisms, some of which might be excessively detailed in the results (e.g., Lines 98–100). Flow and conciseness could still be improved (e.g., the beginning of the paragraph at Line 168; Line 182 might be unnecessary; Lines 199–206 could potentially be reframed to encourage future work rather than focusing on

limitations that are later described as not so critical).

I suggest improving the discussion section to more effectively showcase your important findings.

We have revised the opening of the paragraph at line 168 to enhance the flow and clarity of the text.

Lines 168-169: “The radiative forcing agents presented in this study have different spatial and temporal footprints, which is important to consider.”

Additionally, we have removed the statement on line 182, as we agree it does not contribute meaningfully to the understanding and interpreting of our results.

We have revised lines 199–206 to maintain acknowledgment of the existing limitations, but we now also highlight a key opportunity for future work by steering the discussion toward the need for in situ measurements of permafrost carbon emissions. We have incorporated this on lines 197-199 and 210-212.

Lines 197-199: “Future work should prioritize quantifying post-fire permafrost carbon emissions using in situ measurements, to better constrain post-fire carbon releases from permafrost carbon^{53,54}.”

Lines 210-212: “Future work would benefit from conceptually bringing together carbon dynamics and fluxes from post-fire recovering vegetation and soil respiration from modern plant and aged permafrost carbon.”

53. Estop-Aragonés, C. *et al.* Respiration of aged soil carbon during fall in permafrost peatlands enhanced by active layer deepening following wildfire but limited following thermokarst. *Environmental Research Letters* **13**, (2018).

54. Estop-Aragonés, C. *et al.* Limited release of previously-frozen C and increased new peat formation after thaw in permafrost peatlands. *Soil Biol Biochem* **118**, 115–129 (2018).

Comment 6: Figure 1: It would be helpful to identify the borders of Interior Alaska and Boreal Shield, or alternatively, use slightly different colors (perhaps white/ light grey) for both regions to enhance clarity.

Thank you for your suggestion. We have revised our maps in figure 1 to highlight the boundaries of Interior Alaska and the Boreal Shield better. Interior Alaska is now delineated in blue, while the boreal shield is delineated in orange. This has been applied to all map figures in the manuscript.

Figure 1 | Cumulative mean climate radiative forcing from fires between 2001 and 2019 across Alaska and western Canada over a 70-year period. a, Net radiative forcing map. b, Contribution of net radiative forcing (black circles), direct greenhouse gas and precursor emissions (red squares), permafrost greenhouse gas emissions (orange upside-down triangles), change in surface albedo (blue triangles), aerosol emissions (cyan hexagons), and vegetation recovery (green diamonds) for Interior Alaska and Boreal Shield fires. The error bars indicate the uncertainty of each agent with one standard deviation based on the uncertainty assessment. c, Net radiative forcing over some selected fires in Interior Alaska. d, Net radiative forcing over some selected fires in central Canada. The ecoregion boundary for Interior Alaska is delineated in blue, while the Boreal Shield is delineated in orange.

Comment 7: Fig. 3: For consistency, please revise the use of "Interior Alaska" and "Alaskan Interior" throughout the document.

Thank you for pointing this out. We have updated Figure 3, including its caption, to use "Interior Alaska" and revised the entire manuscript to ensure consistent terminology throughout.

Figure 3 | Radiative forcing from permafrost greenhouse gas emissions induced by fire across landscapes with different permafrost extents. A, Radiative forcing from permafrost greenhouse gas emissions under different permafrost landscapes from all fires across the study domain between 2001 and 2019. B, The net radiative forcing for different permafrost landscapes. Fires from the Interior Alaska are shown in green, Boreal Shield fires in yellow, and domain-wide estimate across western boreal North America in orange. White circles indicate mean values for the different permafrost extents. Permafrost extent was defined using the permafrost zonation index from Gruber⁷⁴. Note, the Boreal Shield ecoregion does not have landscapes with continuous permafrost. Horizontal lines represent the median, the white circle represents the mean and upper and lower limits of the boxes show the 25th and 75th percentiles. Whiskers extend up to 1.5 times the interquartile range.